# TEST-TIME ROBUST PERSONALIZATION FOR FEDERATED LEARNING

**Liangze Jiang**[2,*]**, Tao Lin**[1,*]
liangze.jiang@epfl.ch; lintao@westlake.edu.cn
[1]Research Center for Industries of the Future, Westlake University    [2]EPFL

## ABSTRACT

Federated Learning (FL) is a machine learning paradigm where many clients collaboratively learn a shared global model with decentralized training data. Personalization on FL model additionally adapts the global model to different clients, achieving promising results on consistent local training & test distributions. However, for *real-world* personalized FL applications, it is crucial to go one step further: *robustifying FL models under the evolving local test set during deployment, where various types of distribution shifts can arise*. In this work, we identify the pitfalls of existing works under test-time distribution shifts and propose **Federated Test-time Head Ensemble plus tuning** (**FedTHE+**), which personalizes FL models with robustness to various test-time distribution shifts. We illustrate the advancement of FedTHE+ (and its degraded computationally efficient variant FedTHE) over strong competitors, for training various neural architectures (CNN, ResNet, and Transformer) on CIFAR10 and ImageNet and evaluating on diverse test distributions. Along with this, we build a benchmark for assessing the performance and robustness of personalized FL methods during deployment. Code: https://github.com/LINs-lab/FedTHE.

## 1 INTRODUCTION

Federated Learning (FL) is an emerging ML paradigm that many clients collaboratively learn a shared global model while preserving privacy (McMahan et al., 2017; Lin et al., 2020b; Kairouz et al., 2021; Li et al., 2020a). As a variant, Personalized FL (PFL) adapts the global model to a personalized model for each client, showing promising results on In-Distribution (ID).

However, such successes of PFL *may not* persist during the deployment for FL, as the incoming local test samples are evolving and various types of Out-Of-Distribution (OOD) shifts can occur (compared to local training distribution). Figure 1 showcases some potential test-time distribution shift scenarios, e.g., *label distribution shift* (c.f. (i) & (ii)) and *co-variate shift* (c.f. (iii)): (i) clients may encounter new local test samples in unseen classes; (ii) even if no unseen classes emerge, the local test class distribution may become different; (iii) as another real-world distribution shift, local new test samples may suffer from common corruptions (Hendrycks & Dietterich, 2018) (also called synthetic distribution shift) or natural distribution shifts (Recht et al., 2018). More crucially, the distribution shifts can appear in a mixed manner, i.e. new test samples undergo distinct distribution shifts, making robust FL deployment more challenging.

Making FL more practical requires generalizing FL models to *both* ID & OOD and properly estimating their deployment robustness. To this end, we *first* construct a new PFL benchmark that mimics ID & OOD cases that clients would encounter during deployment, since previous benchmarks cannot measure robustness for FL. Surprisingly, there is a significant discrepancy between the mainstream PFL works and the requirements for real-world FL deployment: existing works (McMahan et al., 2017; Collins et al., 2021; Li et al., 2021b; Chen & Chao, 2022; Deng et al., 2020a) suffer from severe accuracy drop under various distribution shifts, as shown in Figure 2(a).

Test Time Adaptation (TTA)—being orthogonal to training strategies for OOD generalization—has shown great potential in alleviating the test-time distribution shifts. However, as they were designed for homogeneous and non-distributed settings, current TTA methods offer limited performance gains in FL-specific distribution shift scenarios (as shown in Figure 2(b)), especially in the label distribution shift case that is more common and crucial in non-i.i.d. setting.

---

*Equal contribution. Correspondence to: Tao Lin.

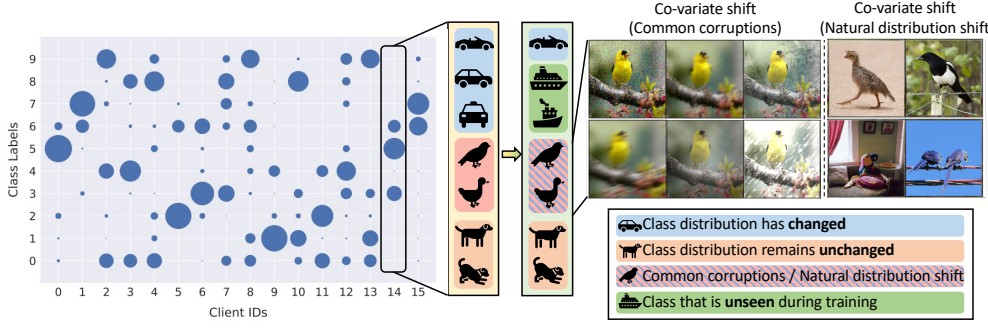

Different scenarios (evolving test distribution) for FL during test time

Figure 1: **Potential distribution shift scenarios for FL during deployment**, e.g., (1) new test samples with unseen labels; (2) class distribution changes within seen labels; (3) new test samples suffer from *co-variate shifts*, either from *common corruptions* or *naturally shifted* datasets. In summary, *Car & Boat*: label distribution shift; *Dog*: unchanged; *Bird*: co-variate shift.

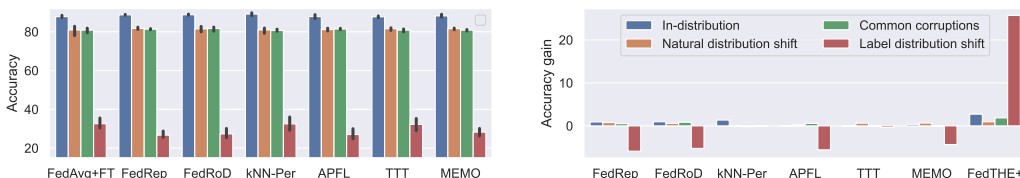

(a) The limitations of PFL & naively adapted TTA methods on test-time distributions shifts in FL.

(b) The accuracy gain of baselines and FedTHE+ (ours) compared to FedAvg+FT on ID & OOD cases.

Figure 2: **Neither existing PFL methods nor applying TTA methods on PFL is sufficient to handle the issues.**

As a remedy and our key contribution, we propose to personalize and robustify the FL model by our computationally efficient Federated Test-time Head Ensemble plus tuning (FedTHE+): we *unsupervisedly and adaptively ensemble* a global generic and local personalized classifiers of a two-head FL model for single test-sample and then conduct an unsupervised test-time fine-tuning. We show that our method significantly improves accuracy and robustness on 1 ID & 4 OOD test distributions, via extensive numerical investigation on strong baselines (including FL, PFL, and TTA methods), models (CNN, ResNet, and Transformer), and datasets (CIFAR10 and ImageNet). Our main contributions are:

- We revisit the evaluation of PFL and identify the crucial test-time distribution shift issues: *a severe gap exists between the current PFL methods in academia and real-world deployment needs.*
- We propose a novel *test-time robust personalization framework (FedTHE/FedTHE+)*, with superior ID & OOD accuracy (throughout baselines, neural networks, datasets, and test distribution shifts).
- As a side product to the community, we provide the *first* PFL benchmark considering a comprehensive list of test distribution shifts and offer the potential to develop realistic and robust PFL methods.

## 2   RELATED WORK

We give a compact related work here due to space limitations. A complete discussion is in Appendix A.

**Federated Learning (FL).** Most FL works focus on facilitating learning under non-i.i.d. client training distribution, leaving the crucial *test-time distribution shift* issue (our focus) unexplored.

**Personalized FL (PFL).** The most straightforward PFL method is to locally fine-tune the global model (Wang et al., 2019; Yu et al., 2020). Apart from these, Deng et al. (2020a); Mansour et al. (2020); Hanzely & Richtárik (2020); Gasanov et al. (2021) linearly interpolate the locally learned model and the global model for evaluation. This idea is extended in Huang et al. (2021); Zhang et al. (2021b) by considering better weight combination strategies over different local models. Inspired by representation learning, Arivazhagan et al. (2019); Collins et al. (2021); Chen & Chao (2022); Tan et al. (2021) suggest decomposing the neural network into a shared feature extractor and a personalized head. FedRoD (Chen & Chao, 2022) uses a similar two-headed network as ours, and explicitly decouples the local and global optimization objectives. pFedHN (Shamsian et al., 2021) and ITU-PFL (Amosy et al., 2021) both use hypernetworks, whereas ITU-PFL focuses on the issue of unlabeled new clients and is orthogonal to our test-time distribution shift setting.

Note that the above-mentioned PFL methods only focus on better generalizing the local training distribution, lacking the resilience to test-time distribution shift issues (our contributions herein).

**OOD generalization in FL.** Investigating OOD generalization for FL is a timely topic. Distribution shifts might occur either geographically or temporally (Koh et al., 2021; Shen et al., 2021). Existing works merely optimize a distributionally robust global model and cannot achieve impressive test-time accuracy. For example, GMA (Tenison et al., 2022) uses a masking strategy on client updates to learn the invariant mechanism across clients. A similar concept can be found in fairness-aware FL (Mohri et al., 2019; Li et al., 2020c; 2021b;a; Du et al., 2021; Shi et al., 2021; Deng et al., 2020b; Wu et al., 2022a), in which accuracy stability for the global model is enforced across local training distributions. However, the real-world common corruptions (Hendrycks & Dietterich, 2019b), natural distribution shifts (Recht et al., 2018; 2019; Taori et al., 2020; Hendrycks et al., 2021b;a) or label distribution shift in FL, is underexplored. We fill this gap and propose FedTHE and FedTHE+ as remedies.

**Benchmarking FL with distribution shift.** Existing FL benchmarks (Ryffel et al., 2018; Caldas et al., 2018; He et al., 2020a; Yuchen Lin et al., 2022; Lai et al., 2022; Wu et al., 2022b) do not properly measure the test accuracy under various ID & OOD cases for PFL. Other OOD benchmarks, Wilds (Koh et al., 2021; Sagawa et al., 2021) and DomainBed (Gulrajani & Lopez-Paz, 2020), primarily concern the generalization over different domains, and hence are not applicable for PFL. Our proposed benchmark for robust FL (BRFL) in section 5 fills this gap, by covering scenarios specific to FL made up of the co-variate and label distribution shift. Note that while the concurrent work (Wu et al., 2022b) similarly argues the sensitivity of PFL to the label distribution shift, they—unlike this work—neither consider other realistic scenarios nor offer a solution.

**Test-Time Adaptation (TTA)** was initially presented in Test-Time Training (TTT) (Sun et al., 2020) for OOD generalization. It utilizes a two-head neural network structure, where the self-supervised auxiliary head/task (rotation prediction) is used to fine-tune the feature extractor. TTT++ (Liu et al., 2021) adds a feature alignment block to Sun et al. (2020), but the fact of requiring accessing the entire test dataset before testing may not be feasible during online deployment. In addition, Tent (Wang et al., 2020a) minimizes the prediction entropy and only updates the Batch Normalization (BN) (Ioffe & Szegedy, 2015) parameters, while MEMO (Zhang et al., 2021a) adapts all model parameters by minimizing the marginal entropy over augmented views. TSA (Zhang et al., 2021c) improves long-tailed recognition via maximizing prediction similarity across augmented views. T3A (Iwasawa & Matsuo, 2021) replaces a trained linear classifier with class prototypes and classifies each sample based on its distance to prototypes. CoTTA (Wang et al., 2022) instead considers a continual scenario, where class-balanced test samples will encounter different corruption types sequentially.

Compared to the existing TTA methods designed for non-distributed and (mostly) iid cases, our two-head approach is uniquely motivated by FL scenarios: we are the first to build effective yet efficient online TTA methods for PFL on heterogeneous training/test local data with FL-specific shifts. Our solution is intrinsically different from TTT variants—rather than using the auxiliary task head to optimize the feature extractor for the prediction task head, our design focuses on two FL-specific heads learned with the global and local objectives and learns the optimal head interpolation weight. Our solution is much better than TSA, as TSA is not suitable for FL scenarios (c.f. Table 3).

## 3 ON THE ADAPTATION STRATEGIES FOR IN- AND OUT-OF- DISTRIBUTION

When optimizing a pre-trained model on a distribution and then assessing it on an unknown test sample that is either sampled from ID or OOD data, there exists a performance trade-off in the choice of adaptation strategies. Our test-time robust personalized FL introduced in section 4 aims to improve OOD accuracy without sacrificing ID accuracy. We illustrate the key intuition below—the two-stage adaptation strategy, inspired by the theoretical justification in Kumar et al. (2022).

Considering a regression task $f_{\mathbf{B},\mathbf{v}}(\mathbf{x}) := \mathbf{v}^\top \mathbf{B} \mathbf{x}$, where feature extractor $\mathbf{B} \in \mathcal{B} = \mathbb{R}^{k \times d}$ is linear and overparameterized, and head $\mathbf{v} \in \mathcal{V} = \mathbb{R}^k$. Let $\mathbf{X} \in \mathbb{R}^{n \times d}$ be a matrix encoding $n$ training examples from the ID data, $\mathcal{S}$ be the $m$-dimensional subspace spanning the training examples, and $\mathbf{Y} \in \mathbb{R}^n$ be the corresponding labels. The following two most prevalent adaptation strategies are investigated: (1) Fine-Tuning (FT), an *effective* scheme for ID that updates all model parameters, and (2) Linear Probing (LP), an *efficient* method where only the last linear layer (i.e. head) is updated. The subscript of $\mathbf{B}$ and $\mathbf{v}$, either FT or LP, indicates the corresponding adaptation strategy.

**Theorem 3.1** (FT can distort pre-trained feature extrator and underperform in OOD (informal version of Kumar et al. (2022))). *Considering FT or LP with training loss $\mathcal{L}(\mathbf{B}, \mathbf{v}) = \left\| \mathbf{X}\mathbf{B}^\top \mathbf{v} - \mathbf{Y} \right\|_2^2$. For FT, the gradient update to $\mathbf{B}$ only happens in the ID space $\mathcal{S}$ and remains unchanged in the orthogonal subspace. The OOD error $\mathcal{L}_{FT}^{OOD}$ of FT iterates $(\mathbf{B}_{FT}, \mathbf{v}_{FT})$, i.e. outside of $\mathcal{S}$, is lower bounded by $\tilde{f}\left(d(\mathbf{v}_0, \mathbf{v}_\star)\right)$, while for LP that iterates $(\mathbf{B}_0, \mathbf{v}_{LP})$, the OOD error $\mathcal{L}_{LP}^{OOD}$ is*

*lower bounded by* $\tilde{f}'(d'(\mathbf{B}_0, \mathbf{B}_\star))$, *where* $\mathbf{v}_0$ *and* $\mathbf{v}_\star$ *correspond to the initial and optimal heads,* $d$ *measures the distance between* $\mathbf{v}_0$ *and* $\mathbf{v}_\star$, *and* $\tilde{f}$ *is a linear transformation function. Similar definitions apply to* $\mathbf{B}_0$, $\mathbf{B}_\star$, $d'$, *and* $\tilde{f}'$.

**Remark 3.2.** *FT and LP present the trade-offs in test performance between ID and OOD data. If the initial feature* $\mathbf{B}_0$ *is close enough to* $\mathbf{B}_\star$, *LP learns a near-optimal linear head with a small OOD error (c.f. lower bound of* $\mathcal{L}_{LP}^{OOD}$*), but FT has a high OOD error. The latter is caused by the coupled gradient updates between* $\mathbf{v}_{FT}$ *and* $\mathbf{B}_{FT}$, *where the distance between* $\mathbf{v}_0$ *and* $\mathbf{v}_\star$ *causes the shifts in* $\mathbf{B}_{FT}$, *and thus leads to distorted features for higher OOD error (c.f. lower bound of* $\mathcal{L}_{FT}^{OOD}$*). If we initialize the head* $\mathbf{v}$ *perfectly at* $\mathbf{v}_\star$, *then FT updates may not increase the OOD error (c.f. lower bound of* $\mathcal{L}_{FT}^{OOD}$*).*

Remark 3.2, as supported by the empirical justifications in subsection E.1, motivates a two-stage adaptation strategy—first performing LP and then FT (a.k.a. LP-FT)—to trade off the performance of LP and FT across ID & OODs. Together with our other unique techniques presented below for PFL scenarios, LP-FT forms the basis of our test-time robust personalized FL method.

## 4 TEST-TIME ROBUST PERSONALIZATION FOR FEDERATED LEARNING

We achieve robust PFL by extending the *supervised* two-stage LP-FT to an *unsupervised* test-time PFL method: (i) *Unsupervised Linear Probing (FedTHE)*: only a scalar head ensemble weight is efficiently optimized for the single test sample to yield much more robust predictions. (ii) *Unsupervised Fine-Tuning*: unsupervised FT (e.g. MEMO) on top of (i) is performed for more accuracy gains and robustness. Stacking them, we propose *Federated Test-time Head Ensemble plus tuning (FedTHE+)*.

### 4.1 PRELIMINARY

The standard FL typically considers a sum-structured optimization problem:

$$\min_{\mathbf{w} \in \mathbb{R}^d} \mathcal{L}(\mathbf{w}) = \sum_{k=1}^K p_k \mathcal{L}_k(\mathbf{w}), \tag{1}$$

where the global objective function $\mathcal{L}(\mathbf{w}) : \mathbb{R}^d \to \mathbb{R}$ is a weighted sum of local objective functions $\mathcal{L}_k(\mathbf{w}) = 1/|\mathcal{D}_k| \sum_{\xi \in \mathcal{D}_k} \ell(\mathbf{x}_\xi, y_\xi; \mathbf{w})$ over $K$ clients. $p_k$ represents the weight of client $k$ normally chosen as $p_k := |\mathcal{D}_k|/\sum_{k=1}^K |\mathcal{D}_k|$, where $|\mathcal{D}_i|$ is the number of samples in local training data $\mathcal{D}_k$. $\ell$ is the sample-wise loss function and $\mathbf{w}$ is the trainable model parameter.

In contrast to traditional FL, which aims to optimize an optimal global model $\mathbf{w}$ across heterogeneous local data distributions $\{\mathcal{D}_1, \ldots, \mathcal{D}_K\}$, personalized FL pursues adapting to each local data distribution $\mathcal{D}_k$ collaboratively (on top of (1)) and individually using a relaxed optimization objective:

$$\min_{(\mathbf{w}_1, \ldots, \mathbf{w}_K) \in \Omega} \mathcal{L}(\Omega) = \sum_{k=1}^K p_k \mathcal{L}_k(\mathbf{w}_k), \tag{2}$$

where client-wise personalized models can be computed and maintained locally.

### 4.2 FEDERATED TEST-TIME HEAD ENSEMBLE (FEDTHE) AS LINEAR PROBING

**Motivation.** Existing PFL methods mostly focus on adapting a per-client personalized model to each local training distribution by optimizing (2). However, the resulting PFL models naturally lose the robustness to label distribution shift (as well as other related shifts), since the global model obtained by optimizing (1) generalizes to all class labels while the personalized model may learn the biased class distribution. Besides, the invariance in the global model learned during FL may be hampered by naive adaptation. Therefore, it is natural to ask: *How to preserve the robustness of the global model while achieving good accuracy on local distribution?*

**Algorithm overview and workflow.** Inspired by the success of representation learning (He et al., 2020b; Chen et al., 2020a; Radford et al., 2021; Wortsman et al., 2021), we propose to use a two-head FL network. The feature extractor, global head, and personalized head are parameterized by $\mathbf{w}^e$, $\mathbf{w}^g$ and $\mathbf{w}^l$ respectively, where the feature representation of input $\mathbf{x}$ is $\mathbf{h} := f_{\mathbf{w}^e}(\mathbf{x}) \in \mathbb{R}^d$. $\hat{\mathbf{y}}^g := f_{\mathbf{w}^g}(\mathbf{h})$ and $\hat{\mathbf{y}}^l = f_{\mathbf{w}^l}(\mathbf{h})$ denote generic and personalized prediction logits. During test-time, the model's prediction $\hat{\mathbf{y}} = e\hat{\mathbf{y}}^g + (1-e)\hat{\mathbf{y}}^l$, where $e \in [0, 1]$ is a *learnable* head ensemble weight. Our training and test-time robust personalization are depicted in Figure 3; the detailed algorithms are in Appendix B.

- **Federated Training.** Similar to the standard FedAvg (McMahan et al., 2017), each client will train the received global model in each round on local training data and then send it back to the server, for the purpose of federated learning. Then the personalized head is trained with a frozen and unadapted feature extractor, and will be locally kept. Note that the general TTT methods (Sun et al., 2020; Liu et al., 2021) require the joint training of main and auxiliary task heads for the later test-time adaptation, while we only maintain and adapt the main task head (i.e. no auxiliary task head).

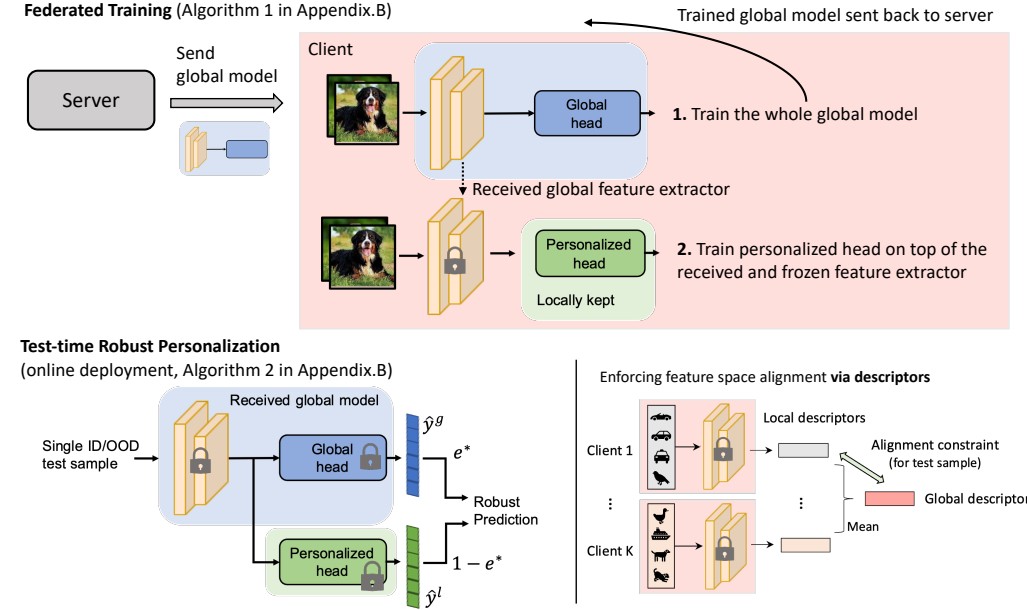

Figure 3: **The training and test phase of FedTHE.** *Top*: learning global and personalized head disentangledly. *Bottom Left*: improving ID & OOD accuracy during online deployment via Test-time Head Enesmble (THE). *Bottom Right*: enforcing feature space alignment to combat overconfident and biased personalized head in FL.

- **Test-time Robust Personalization**. To deal with arbitrary ID & OODs during deployment, *given a single test sample, we unsupervisedly and adaptively interpolate the personalized head and global head through optimizing $e$, as unsupervised Linear Probing*. Such unsupervised optimization for $e$ (while freezing the feature extractor) is illustrated below, which is intrinsically different from the general TTT methods that the feature extractor will be adapted (while freezing their two heads).

**Reducing prediction uncertainty.** Entropy Minimization (EM) is a widely used approach in self-supervised learning (Grandvalet & Bengio, 2004) and TTA (Wang et al., 2020a; Zhang et al., 2021a) for sharpening model prediction and enforcing prediction confidence. Similarly, we minimize the Shannon Entropy (Shannon, 2001) of the two-head weighted logits prediction $\hat{\mathbf{y}}$ as our first step:

$$\mathcal{L}_{\text{EM}} = -\sum_i p_i(\hat{\mathbf{y}}) \log p_i(\hat{\mathbf{y}}) \,, \tag{3}$$

where $p_i(\hat{\mathbf{y}})$ represents the softmax probability of $i$-th class on $\hat{\mathbf{y}}$, and $\mathbf{w}^g$ and $\mathbf{w}^l$ are frozen. In practice, we apply a softmax operation over two learnable scalars. However, while optimizing $e$ by (3) improves the prediction confidence, several limitations may arise in FL. Firstly, higher prediction confidence brought by EM does not always indicate improved accuracy. Secondly, the personalized head learned from the local training data can easily overfit to the corresponding biased class distribution, and such overconfidence also hinders the adaptation quality and reliability to arbitrary distribution shifts.

**Enforcing feature space alignment.** As a countermeasure to EM's limitations, we apply feature space constraints to optimize the head interpolation weight. Such strategy stabilizes the adaptation by leveraging the FL-specific guideline from the local client space and global space (of the FL system), where minimizing a Feature Alignment (FA) loss allocates higher weight to the head (either local personalized or global generic) that has a closer alignment to the test feature.

Specifically, three feature descriptors (2 from training and 1 in the test phase) are maintained, as shown in Figure 3 (downright): per-client local descriptor $\mathbf{h}^l$, global descriptor $\mathbf{h}^g$, and test history descriptor $\mathbf{h}^{\text{history}}$. *All of them are computed from the global feature extractor received from the server.* ● local descriptor $\mathbf{h}^l$ is an average of local training samples' feature representations. ● During federated training, each client forwards its $\mathbf{h}^l$ to the server along with the trained global model parameters (i.e. $\mathbf{w}^e$ and $\mathbf{w}^g$), and the server generates the global descriptor $\mathbf{h}^g$ via averaging all local descriptors. The global descriptor will be synchronized with sampled clients, along with the global model, for the next communication round. ● To reduce the instability of the high variance in a single test feature, we explore a test history descriptor maintained by the Exponential Moving Average (EMA) during deployment $\mathbf{h}^{\text{history}} := \alpha \mathbf{h}_{n-1} + (1-\alpha) \mathbf{h}^{\text{history}}$, which is initialized as the feature of the first test sample.

The FA loss then optimizes a weighted distance of the test feature to global and local features:

$$\mathcal{L}_{\text{FA}} = e \left\| \mathbf{h}_n^{'} - \mathbf{h}^g \right\|_2 + (1 - e) \left\| \mathbf{h}_n^{'} - \mathbf{h}^l \right\|_2 , \tag{4}$$

where $\mathbf{h}_n^{'} := \beta \mathbf{h}_n + (1 - \beta) \mathbf{h}^{\text{history}}$ represents a smoothed test feature, $\|\cdot\|_2$ is the euclidean norm, and $e$ indicates the *learnable* ensemble weight (same as EM). Note that $h_n^{'}$ is only used for computing $\mathcal{L}_{\text{FA}}$ (not used for prediction), and we use constant $\alpha = 0.1$ and $\beta = 0.3$ throughout the experiments[1]. Combining (3) and (4) works as an arms race on classifier and feature extractor perspective, which is simple yet effective in various experimental settings (c.f. section 6), including neural architectures, datasets, and distribution shifts. Note that although other FA strategies such as Gretton et al. (2012); Lee et al. (2018); Ren et al. (2021) might be more advanced, they are not specifically designed for FL, and adapting them to FL is non-trivial. We leave these to future work.

**Similarity-guided Loss Weighting (SLW).** Overall, naively optimizing the sum of (3) and (4) is unlikely to reach the optimal test-time performance. While numerous works introduce extra hyper-parameters/temperatures to weight the loss terms, such a strategy is prohibitively costly and inapplicable during *online deployment*. To save tuning effort, we form an adaptive unsupervised loss $\mathcal{L}_{\text{SLW}}$ with the cosine similarity $\lambda_s$ of the probability outputs from the global and personalized heads:

$$e^{\star} = \arg\min_e \left( \mathcal{L}_{\text{SLW}} := \lambda_s \mathcal{L}_{\text{EM}} + (1 - \lambda_s) \mathcal{L}_{\text{FA}} \right) , \quad \text{where } \lambda_s = \cos\left( p(\hat{\mathbf{y}}^g), p(\hat{\mathbf{y}}^l) \right) \in [0, 1] . \tag{5}$$

Therefore, for two similar logit predictions (i.e. large $\lambda_s$) from two heads, a more confident head is preferred, while for two dissimilar predictions (i.e. small $\lambda_s$), feature alignment loss regularizes the issue of over-fitting, and assists to attain equilibrium between prediction confidence and head expertise.

**Discussions.** Generally, FedTHE exploits the potential robustness of combining the federated learned global head and local personalized head. To the best of our knowledge, we are the first to design the FL-specific TTA method: such a method can not only handle co-variate distribution shift, but also other shift types coupled with label distribution shift which are crucial in FL. Other methods for improving global model quality (Li et al., 2020b; Wang et al., 2020b; Karimireddy et al., 2020) are orthogonal[2] and compatible[3] with ours. Our method is also suitable for generalizing new clients[4]. Besides, our method requires marginal hyperparameter tuning efforts (as shown in Appendix E.4), with guaranteed efficiency: only a personalized head is trained during the personalization phase and only a scalar ensemble weight $e$ is optimized for test samples during the test-time personalization phase.

### 4.3 FEDTHE+: TEST-TIME HEAD ENSEMBLE PLUS TUNING FOR ROBUST PERSONALIZED FL

Following the insights stated in section 3, we further employ MEMO (Zhang et al., 2021a) as our test-time FT phase, formulating our LP-FT based test-time robust PFL method, termed FedTHE+. The test-time FT updates the network parameters (i.e. $\mathbf{w}^e$, $\mathbf{w}^g$, and $\mathbf{w}^l$) while fixing the optimized head ensemble weight $e^{\star}$. Pseudocodes are deferred to Appendix B. As we will show in section 6, applying FedTHE is already sufficient to produce improved ID & OOD accuracy compared to various baselines, while FedTHE+ achieves more accuracy and robustness gains and benefits from the LP-FT scheme.

## 5 BRFL: A BENCHMARK FOR ROBUST FL

As there is no existing work that properly evaluates the robustness of FL, we introduce Benchmark for Robust Federated Learning (BRFL), as illustrated in Figure 4 with CIFAR10 as an example.

**Clients with heterogeneous data.** Following the common idea in Yurochkin et al. (2019); Hsu et al. (2019); He et al. (2020a), we use the Dirichlet distribution to create heterogeneous clients with disjoint non-i.i.d. data that is never shuffled across clients. The smaller degree of heterogeneity, the more likely the clients hold examples from fewer classes. We then uniformly partition each client's data into local training/validation/test sets.

---

[1]We also investigate how different choice of $\alpha$ and $\beta$ impact the performance of FedTHE in Appendix E.4.

[2]Merely applying BalancedSoftmax to PFL is far from handling test-time distribution shift (c.f. Table 13).

[3]As a side study, similar to Chen & Chao (2022), we use Balanced Risk Minimization (i.e. BalancedSoftmax (Ren et al., 2020)) during local training for a better global model, justified by the improvements in Table 3.

[4]Local training feature $\mathbf{h}^l$ only relies on the received feature extractor and local training/personalization data, shared global feature $\mathbf{h}^g$ is synchronized with model parameters, and test history feature is only initialized locally.

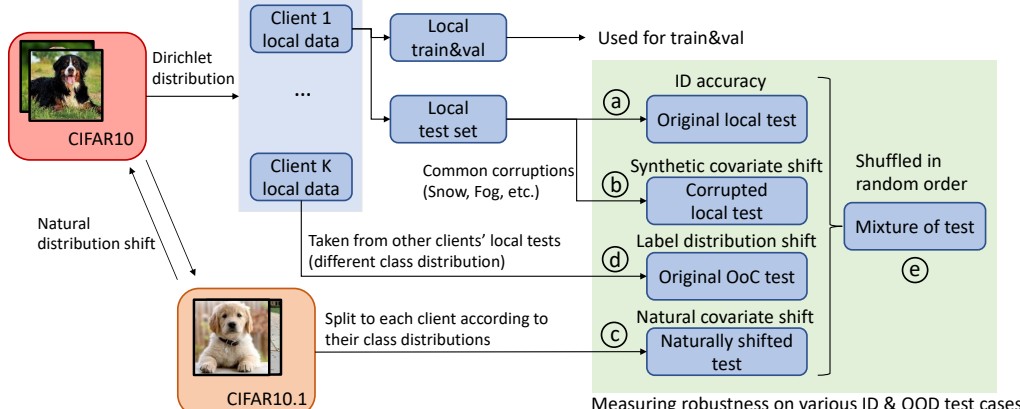

Figure 4: **The data pipeline of our benchmark (BRFL) for evaluating FL robustness.**

**Various test distributions.** We construct 5 distinct test distributions, including 1 ID and 4 OODs. The ID test is the original local test, which is i.i.d. with local training distribution. And the OODs are shifted from local training distribution, covering the two most common OOD cases encountered in FL, i.e. co-variate shift and label distribution shift. The examined test distributions are termed as: ⓐ Original local test, ⓑ Corrupted local test, ⓒ Naturally shifted test (same distribution as in ⓐ), ⓓ Original Out-of-Client (OoC) local test, and ⓔ Mixture of test (from ⓐ, ⓑ, ⓒ, and ⓓ). Specifically, ⓐ reflects how effectively the personalized model adapts to in-distribution, whereas ⓑ and ⓒ represent co-variate shift and ⓓ investigates label distribution shift by sampling from other clients' test data. However, each of ⓐ – ⓓ merely considers a single type of real-world test distributions, thus we build ⓔ for a more realistic test scenario, by randomly drawing test samples from the previous 4 test distributions.

**Datasets.** We consider CIFAR10 (Krizhevsky et al., 2009) and ImageNet (Deng et al., 2009) (down-sampled to the resolution of 32 (Chrabaszcz et al., 2017) in our case due to computational infeasibility); future work includes adapting other datasets (Koh et al., 2021; Sagawa et al., 2021; Gulrajani & Lopez-Paz, 2020). For CIFAR10, we construct the synthetic co-variate shift ⓑ by leveraging 15 common corruptions (e.g. Gaussian Noise, Fog, Blur) in CIFAR10-C[5] (Hendrycks & Dietterich, 2018), and split CIFAR10.1 (Recht et al., 2018) for natural co-variate shift ⓐ. For ImageNet[6], natural co-variate shifts ⓒ are built by splitting ImageNet-A (Hendrycks et al., 2021b), ImageNet-R (Hendrycks et al., 2021a), and ImageNet-V2 (Recht et al., 2019) (MatchedFrequency test) to each client based on its local class distribution. The detailed dataset construction procedure is in Appendix D.

## 6 EXPERIMENTS

With the proposed benchmark BRFL, we evaluate our method and the strong baselines on various neural architectures, datasets, and test distributions (1 ID and 4 OODs). *FedTHE and FedTHE+ exhibit consistent and noticeable performance gains over all competitors during test-time deployment.*

### 6.1 SETUP

We outline the evaluation setup for test-time personalized FL; for more details please refer to Appendix C. We consider the classification task, a standard task considered by all FL methods. In all our FL experiments, results are averaged across clients and each setting is evaluated over 3 seeds. **Models.** To avoid the pitfalls caused by BN layers in FL (Li et al., 2021c; Diao et al., 2021) (orthogonal to our contributions), we start with a simple CNN architecture (LeCun et al., 1998; Chen & Chao, 2022) (w/o BN layers), and then extend to ResNet20 (He et al., 2016) (w/ GN (Hsieh et al., 2020)) and Compact Convolutional Transformer (Hassani et al., 2021) (a computational feasible Transformer). **Strong Baselines** from FL, PFL, or TTA (naively used in FL), are taken into account:

- **FL methods**[7]: FedAvg (McMahan et al., 2017), GMA (Tenison et al., 2022): learn a global model.

---

[5]For each test sample, we randomly sample a corruption and apply it to the sample. This makes the test set more challenging and realistic compared to original CIFAR10-C since the corruptions come in random order.

[6]The overlapped 86 classes Hendrycks et al. (2021b;a); Recht et al. (2019) are used to ensure consistent class distribution between ID and OOD.

[7]Our approaches are orthogonal to efforts to improve training quality of global (and thus personalized) model.

- **Strong PFL methods**: (i & ii) FedRep (Collins et al., 2021) and FedRoD (Chen & Chao, 2022) that similarly use a decoupled feature extractor & classifier(s); (iii) APFL (Deng et al., 2020a), a weighted ensemble of personalized and global models; (iv) Ditto (Li et al., 2021b), a fairness-aware PFL method that outperforms other fairness FL methods TERM (Li et al., 2020c; 2021a) and Per-FedAvg (Fallah et al., 2020); (v) FedAvg + FT, an effective choice that fine-tunes on the global model on local training data; and (vi) kNN-Per (Marfoq et al., 2022), a SOTA method that interpolates a global model with a local k-nearest neighbors (kNN) model.
- **TTA methods**: TTT (online version) (Sun et al., 2020), MEMO (Zhang et al., 2021a), and T3A (Iwasawa & Matsuo, 2021). We adapt them to personalized FL[8] by adding test-time adaptation to the FedAvg + FT personalized model, and prediction on each test point occurs immediately after adaptation[9]. We also apply MEMO to the global model given the good performance of MEMO (P) on the FedAvg + FT model, termed MEMO (G). We omit the comparison with TENT (Wang et al., 2020a) as it relies on BN and underperforms MEMO (Zhang et al., 2021a).

## 6.2 RESULTS

Table 1: **Test accuracy of baselines across various test distributions and different degrees of heterogeneity.** A simple CNN is trained on CIFAR10. The client heterogeneity is determined by the value of Dirichlet distribution (Yurochkin et al., 2019; Hsu et al., 2019), termed "Dir". **The left inline figure** illustrates the learning curves of methods, evaluated on the "mixture of test" with Dir(0.1). More details refers to Appendix E.2. **The right inline figure** shows the detailed comparison between ours and strong baselines across 1 ID and 4 OODs.

| Methods | Local data distribution: Dir(0.1) | | | | | Local data distribution: Dir(1.0) | | | | |
|---|---|---|---|---|---|---|---|---|---|---|
| | Original local test | Corrupted local test | Original OoC local test | Naturally Shifted test | Mixture of test | Original local test | Corrupted local test | Original OoC local test | Naturally Shifted test | Mixture of test |
| FedAvg | 64.25 ±1.42 | 52.09 ±1.53 | 64.31 ±1.77 | 50.09 ±1.36 | 57.68 ±1.49 | 73.23 ±0.38 | 56.99 ±1.16 | 73.13 ±0.13 | 57.17 ±0.34 | 64.97 ±0.42 |
| GMA | 63.40 ±1.22 | 51.40 ±1.30 | 63.53 ±1.48 | 49.57 ±0.87 | 56.98 ±1.20 | 72.25 ±0.40 | 56.87 ±0.85 | 72.11 ±1.07 | 56.49 ±0.42 | 64.43 ±0.40 |
| MEMO (G) | 67.51 ±1.59 | 53.96 ±1.68 | **66.85** ±1.17 | 53.61 ±1.21 | 60.48 ±1.39 | 74.34 ±1.09 | 57.71 ±1.07 | **74.81** ±0.76 | 59.46 ±0.27 | 66.58 ±0.75 |
| FedAvg + FT | 87.82 ±0.59 | 80.85 ±1.24 | 32.59 ±2.69 | 80.95 ±2.55 | 70.55 ±0.45 | 78.21 ±0.68 | 65.70 ±0.60 | 60.01 ±0.92 | 65.59 ±1.46 | 67.38 ±0.41 |
| APFL | 87.85 ±1.01 | 81.43 ±0.35 | 27.02 ±2.71 | 81.15 ±0.74 | 69.36 ±0.74 | 68.73 ±1.93 | 59.25 ±0.98 | 57.30 ±3.09 | 31.46 ±1.03 | 55.74 ±1.76 |
| FedRep | 88.73 ±0.26 | 81.37 ±0.30 | 26.70 ±1.85 | 81.72 ±0.63 | 69.63 ±0.36 | 79.25 ±0.59 | 65.87 ±0.65 | 54.21 ±1.58 | 66.69 ±1.22 | 66.50 ±0.21 |
| Ditto | 86.55 ±1.04 | 79.39 ±1.77 | 22.46 ±2.80 | 77.61 ±2.02 | 66.51 ±0.37 | 71.05 ±1.13 | 60.82 ±0.81 | 39.40 ±0.37 | 56.91 ±2.64 | 57.05 ±1.11 |
| FedRoD | 88.78 ±0.32 | 81.69 ±1.18 | 27.35 ±2.50 | 81.52 ±1.43 | 69.84 ±0.25 | 78.55 ±1.42 | 66.03 ±1.48 | 54.94 ±2.04 | 65.47 ±2.32 | 66.25 ±0.93 |
| kNN-Per | 89.18 ±0.69 | 80.67 ±0.61 | 32.50 ±3.45 | 80.98 ±1.53 | 70.83 ±0.77 | 79.66 ±0.93 | 64.19 ±0.85 | 64.38 ±2.23 | 66.86 ±2.74 | 68.78 ±0.55 |
| TTT | 87.73 ±0.68 | 80.88 ±0.81 | 32.22 ±3.13 | 81.59 ±0.77 | 70.60 ±0.66 | 77.64 ±0.49 | 65.15 ±0.73 | 58.68 ±1.49 | 65.34 ±1.15 | 66.70 ±0.32 |
| MEMO (P) | 88.19 ±0.79 | 80.87 ±0.44 | 28.25 ±1.96 | 81.61 ±0.51 | 69.73 ±0.46 | 80.20 ±0.69 | 66.86 ±0.82 | 57.80 ±2.44 | **68.79** ±1.46 | 68.41 ±0.50 |
| T3A | 83.78 ±1.73 | 75.24 ±2.16 | 36.76 ±2.71 | 79.17 ±2.38 | 68.74 ±1.13 | 73.80 ±0.63 | 59.58 ±0.63 | 60.83 ±0.72 | 64.87 ±1.42 | 64.77 ±0.38 |
| FedTHE (Ours) | 89.58 ±0.45 | 81.62 ±0.93 | 57.12 ±3.66 | 80.88 ±0.80 | 72.07 ±0.57 | 80.49 ±0.89 | 66.93 ±1.03 | 68.39 ±0.64 | 66.25 ±0.96 | 69.88 ±1.19 |
| FedTHE+ (Ours) | **90.55** ±0.41 | **82.71** ±0.69 | 58.29 ±2.05 | **81.91** ±0.54 | **72.90** ±0.93 | **82.05** ±0.69 | **68.75** ±0.73 | 69.49 ±0.84 | **68.45** ±1.61 | **71.59** ±0.40 |

Table 2: **Test accuracy of baselines on different ImageNet test distributions.** ResNet20-GN with width factor of 4 is used for training ImageNet on clients with heterogeneity Dir(1), and results on Dir(0.1) can be found in Appendix E.3. Our methods are compared with 5 strong competitors on 1 ID data and 4 OOD data (partitioned based on section 5), and report the mean/std over these 5 test scenarios.

| Methods | Local test of ImageNet | ImageNet-A | ImageNet-V2 | ImageNet-R | Mixture of test | Average |
|---|---|---|---|---|---|---|
| FedAvg + FT | 69.34 ±0.51 | 8.08 ±0.45 | 21.36 ±0.36 | 55.81 ±5.00 | 38.65 ±1.31 | 38.65 ±24.9 |
| FedRoD | 70.52 ±2.53 | 7.17 ±0.44 | 19.11 ±0.14 | 52.39 ±4.79 | 37.30 ±1.72 | 37.30 ±25.4 |
| FedRep | 62.00 ±1.69 | 7.07 ±0.54 | 19.08 ±0.16 | 52.63 ±0.83 | 35.19 ±0.46 | 35.19 ±22.8 |
| MEMO (P) | 71.89 ±0.96 | 8.21 ±0.45 | 22.29 ±0.40 | 56.95 ±3.25 | 39.83 ±1.13 | 39.83 ±25.6 |
| T3A | 68.78 ±1.06 | **8.26** ±0.65 | 21.39 ±0.33 | 56.80 ±2.64 | 38.81 ±0.67 | 38.81 ±24.8 |
| FedTHE (Ours) | 86.56 ±0.64 | 6.76 ±0.40 | 25.26 ±0.19 | 59.62 ±4.43 | 41.41 ±1.63 | 43.92 ±30.8 |
| FedTHE+ (Ours) | **88.51** ±0.79 | 6.80 ±0.65 | **26.34** ±0.49 | **60.34** ±3.49 | **42.39** ±0.86 | **44.88** ±31.4 |

**Superior ID&OOD accuracy of our approaches across diverse test scenarios.** We compare FedTHE and FedTHE+ with extensive baselines in Table 1, on different ID & OODs of CIFAR10 and different non-i.i.d. degrees. *Our approaches enable consistent and significant ID accuracy and OOD robustness gains (across test distributions, competitors, and different data heterogeneity degrees. They also accelerate the optimization (e.g. by 1.5×–2.5×)* as shown in Figure 12 and Figure 13 in Appendix. More evaluations (including entangled distribution shifts) refer to Appendix E.2 and E.3.

**Generalizability on other models and datasets.** Figure 5 summarizes the accuracy of strong baselines on SOTA models (i.e. ResNet20 with GN (Wu & He, 2018) and CCT4 with LN (Ba et al., 2016)), while Table 2 examines the challenging ImageNet. *The benefits of our approaches are further pronounced in both cases on all ID & OOD test scenarios (compared to Table 1), achieving at least a 10% increase in test accuracy for most test scenarios.* On the downside, the unsatisfactory results

---

[8]As justified in Table 1, FedAvg + FT is a very strong competitor over other personalized FL methods, and we believe it is sufficient to examine these test-time methods on the model from FedAvg + FT.

[9]Sample-wise T3A performs similarly to batch-wise (thus sample-wise T3A is used for fair comparisons).

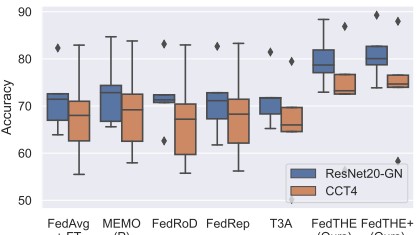

Figure 5: **Test accuracy of baselines across different architectures** (training ResNet20-GN and CCT4 on CIFAR10 with heterogeneity Dir(1.0)). We compare our methods with five strong competitors and report the mean/std of results over five test distributions (1 ID and 4 OODs): each result is evaluated on one test distribution/scenario (average over three different seeds) during online deployment. Detailed settings defer to Appendix C. Numerical results on different distribution shifts can be found in Table 6 (Appendix E.3).

Table 3: **Ablation study for different design choices of FedTHE+** (training CNN on CIFAR10 with Dir(0.1)). The indentation with different symbols denotes adding (+) / removing (−) a component, or using a variant (◦).

| Design choices | Methods | Original local test | Corrupted local test | Original OoC local test | Naturally Shifted test | Mixture of test | Average |
|---|---|---|---|---|---|---|---|
| ① | MEMO (P): a strong baseline | $88.19_{\pm0.79}$ | $80.87_{\pm0.44}$ | $28.25_{\pm1.96}$ | $81.61_{\pm0.51}$ | $69.73_{\pm0.46}$ | $69.73_{\pm24.1}$ |
| ② | + test-time LP-FT | $88.52_{\pm0.41}$ | $81.51_{\pm0.83}$ | $28.62_{\pm2.63}$ | $81.75_{\pm0.81}$ | $70.10_{\pm0.35}$ | $70.10_{\pm21.6}$ |
| ③ | TSA on our pure 2-head model | $87.68_{\pm0.62}$ | $79.38_{\pm0.87}$ | $32.38_{\pm1.70}$ | $79.21_{\pm1.95}$ | $69.66_{\pm0.44}$ | $69.66_{\pm21.8}$ |
| ④ | Our pure 2-head model | $87.49_{\pm0.17}$ | $79.74_{\pm0.70}$ | $33.34_{\pm2.46}$ | $79.17_{\pm1.36}$ | $69.90_{\pm0.15}$ | $69.90_{\pm21.4}$ |
| ⑤ | + BalancedSoftmax | $88.68_{\pm0.17}$ | $81.11_{\pm0.59}$ | $32.92_{\pm2.55}$ | $80.18_{\pm1.00}$ | $70.72_{\pm0.36}$ | $70.72_{\pm22.1}$ |
| ⑥ | + FA | $87.41_{\pm0.35}$ | $76.10_{\pm1.12}$ | $63.39_{\pm2.87}$ | $77.00_{\pm1.47}$ | $67.98_{\pm0.72}$ | $74.38_{\pm9.2}$ |
| ⑦ | + EM | $71.37_{\pm1.15}$ | $60.54_{\pm0.66}$ | $49.62_{\pm0.45}$ | $57.38_{\pm1.40}$ | $59.98_{\pm1.23}$ | $59.78_{\pm7.8}$ |
| ⑧ | + Both (i.e. 0.5 FA + 0.5 EM) | $88.54_{\pm0.43}$ | $81.40_{\pm0.47}$ | $41.45_{\pm2.51}$ | $80.55_{\pm0.57}$ | $70.57_{\pm0.79}$ | $72.50_{\pm18.5}$ |
| ⑨ | + SLW (i.e. FedTHE) | $89.58_{\pm0.45}$ | $81.62_{\pm0.93}$ | $57.12_{\pm3.66}$ | $80.88_{\pm0.80}$ | $72.07_{\pm0.57}$ | $76.25_{\pm12.4}$ |
| ⑩ | ◦ Batch-wise FedTHE | $89.97_{\pm0.24}$ | $82.58_{\pm1.21}$ | $54.12_{\pm1.79}$ | $81.58_{\pm1.06}$ | $70.10_{\pm0.26}$ | $75.67_{\pm14.0}$ |
| ⑪ | − Test history $h^{history}$ | $87.05_{\pm0.68}$ | $77.79_{\pm1.23}$ | $50.45_{\pm2.10}$ | $76.99_{\pm0.98}$ | $71.89_{\pm1.28}$ | $72.83_{\pm13.7}$ |
| ⑫ | − BalancedSoftmax | $89.29_{\pm0.75}$ | $81.32_{\pm0.18}$ | $56.46_{\pm1.84}$ | $80.65_{\pm1.08}$ | $71.81_{\pm0.95}$ | $75.91_{\pm12.5}$ |
| ⑬ | + FT (i.e. FedTHE+) | $\mathbf{90.55}_{\pm0.41}$ | $\mathbf{82.71}_{\pm0.69}$ | $58.29_{\pm2.05}$ | $\mathbf{81.91}_{\pm0.54}$ | $\mathbf{72.90}_{\pm0.93}$ | $\mathbf{77.27}_{\pm12.3}$ |
| ⑭ | − BalancedSoftmax | $90.32_{\pm0.69}$ | $82.86_{\pm0.90}$ | $58.19_{\pm3.81}$ | $81.74_{\pm0.53}$ | $72.33_{\pm1.22}$ | $77.09_{\pm12.4}$ |

on ImageNet-A reveal that natural adversarial examples are too challenging for all methods in the field (Hendrycks et al., 2021b) and we leave the corresponding exploration for future work.

**Remarks on the ID&OOD accuracy trade-off for FL.** Being aligned with Miller et al. (2021), the OOD accuracy on corrupted and naturally shifted test samples is positively correlated with ID accuracy. However, for OoC local test case: the existing works for PFL perform poorly under such label distribution shift with tailed or unseen classes, due to overfitting the global model to the local ID training distribution (the better the ID accuracy, the worse the label distribution shift accuracy). The classic FL methods without personalization (such as FedAvg) utilize a global model and achieve considerable accuracy on the OoC case, but they fail to reach high ID accuracy. Also, naively adapting TTA methods to FL or PFL *cannot* produce satisfying ID and co-variate shift accuracy FL methods and *cannot* handle label distribution shift when using on PFL. *Our method finds a good trade-off point by adaptively ensemble the global and personalized head, significantly improving the accuracy in all cases.*

**Ablation study (Table 3).** We ablate FedTHE+ step by step, with EM, FA, SLW, test-time FT, etc.
- Only adding the EM module performs poorly (c.f. ⑦) as the fake confidence stated in section 4.2, while merely adding FA only produces much better OoC local test accuracy (c.f. ⑥). Specifically, FA gives higher weight to the global head (which generalizes better to all class labels) when facing tailed or unseen class samples, since the test feature is more aligned with the global descriptor;
- Synergy exists when combing FA with EM (c.f. ⑧), and SLW (c.f. ⑨) is another remedy;
- Further adding test-time FT phase (c.f. ⑬ and ⑨) improves all cases (the effectiveness of LP-FT in FL is justified), giving much more gains than naively building LP-FT with MEMO (c.f. ② and ①);
- Despite requiring a test history descriptor $h^{history}$, our approaches are robust to the shift types and orders of test samples, as shown in the "Mixture of test" column (c.f. Table 1, Table 2, and Table 3);
- The role of $h^{history}$: when it's removed from FA, accuracy drops greatly (c.f. ⑨, ⑪);
- Our methods outperform TSA (Zhang et al., 2021c) (exploits head aggregation), c.f. ③, ⑨, and ⑬;
- A better global model, while not our focus, gives better results (c.f. ⑨ v.s. ⑫, and ⑬ v.s. ⑭);
- In case of inference efficiency concern, the performance of batch-wise FedTHE is on par with sample-wise FedTHE (c.f. ⑨ and ⑩), exceeding existing strong competitors.

We defer experiments, regarding the model interpolation strategies, a larger number of clients (e.g. 100 clients), number of training epochs, size of the local dataset, number of clients, etc, to Appendix E.4.

**Conclusion.** We *first* identify the importance of generalizing FL models on OODs and build a benchmark for FL robustness. We propose FedTHE+ for test-time (online) robust personalization based on *unsupervised* LP-FT and TTA, showing superior ID & OOD accuracy. Future works include designing more powerful feature alignment for FL and adapting new datasets to the benchmark.

## ACKNOWLEDGEMENT

We thank Martin Jaggi for his comments and support. We also thank the anonymous reviewers and Soumajit Majumder (soumajit.majumder@huawei.com) for their constructive and helpful reviews. This work was supported in part by the Science and Technology Innovation 2030 – Major Project (No. 2022ZD0115100), the Research Center for Industries of the Future (RCIF) at Westlake University, and Westlake Education Foundation.

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

# Supplementary Material

## CONTENTS OF APPENDIX

In appendix, we provide more details and results omitted in the main paper.

- Appendix A offers a full list of related works.
- Appendix B elaborates the training and test algorithms for FedTHE+ and its degraded version FedTHE (c.f. section 4 of the main paper).
- Appendix C provides the implementation details and experimental setups (c.f. subsection 6.2 of the main paper).
- Appendix D includes additional details of constructing various test distributions based on our benchmark for Personalized FL (cf. section 5 of the main paper).
- Appendix E provides more results on different degrees of non-i.i.d local distributions, a more challenging test case, and ablation study (cf. subsection 6.2 of the main paper).

## A  COMPLETE RELATED WORKS

**Federated Learning (FL).** While communicating efficiently, the nature of heterogeneous clients in FL impedes learning performance considerably (Li et al., 2020b; Wang et al., 2020b; Mitra et al., 2021; Diao et al., 2021; Wang et al., 2020b; Lin et al., 2020a; Karimireddy et al., 2020; 2021; Guo et al., 2021). To address such issue of non-i.i.d. client training distributions for federated optimization, some works on proximal regularization (Li et al., 2020b; Wang et al., 2020b), variance reduction (Karimireddy et al., 2020; Haddadpour et al., 2021; Mitra et al., 2021; Karimireddy et al., 2021), and other techniques like momentum (Wang et al., 2020c; Hsu et al., 2019; Reddi et al., 2021; Karimireddy et al., 2021) and clustering (Ghosh et al., 2020; Zhu et al., 2022) were proposed. Despite being promising, most of them focus on better optimizing a global model across clients, leaving the crucial *test-time distribution shift* issue (our focus) under-explored.

**Personalized FL (PFL)** is a natural strategy to trade off the challenges from training a global model with heterogeneous client data and performance requirements of test-time deployment per client (Wang et al., 2019; Yu et al., 2020; Ghosh et al., 2020; Sattler et al., 2020; Chen et al., 2018; Jiang et al., 2019; Singhal et al., 2021; T Dinh et al., 2020; Fallah et al., 2020; Smith et al., 2017; Li et al., 2021b; Tan et al., 2022). In the spirit of this, different from naively fine-tune the global model locally (Wang et al., 2019; Yu et al., 2020), Deng et al. (2020a); Mansour et al. (2020); Hanzely & Richtárik (2020); Gasanov et al. (2021) achieve personalization by linearly interpolating locally learnt model and the global model and use the interpolated model for evaluation. This idea is further extended in Huang et al. (2021); Zhang et al. (2021b) by considering better weight combination

strategies over different local models. As inspired by representation learning, another series of approaches (Arivazhagan et al., 2019; Collins et al., 2021; Chen & Chao, 2022; Tan et al., 2021) suggest decomposing the neural network layers into a shared feature extractor and a personalized head. FedRoD (Chen & Chao, 2022) shares a similar neural architecture as ours, where a two-head network is proposed to explicitly decouple the model's local and global optimization objectives. pFedHN (Shamsian et al., 2021) and ITU-PFL (Amosy et al., 2021) use hypernetworks for personalization, where ITU-PFL aims to solve unlabeled new clients problem, as orthogonal to our setting. Other promising methodologies are also adapted to personalized FL, including meta learning (Chen et al., 2018; Jiang et al., 2019; Singhal et al., 2021; T Dinh et al., 2020; Fallah et al., 2020), multi-task learning (Smith et al., 2017; Sattler et al., 2020; Li et al., 2021b), and others (Achituve et al., 2021).

Note that the mentioned personalized FL methods only focus on better generalizing the local training distribution, lacking of the resilience to test-time distribution shift issues (our contributions herein).

**OOD generalization and its status in FL.** OOD generalization is a longstanding problem (Hand, 2006; Quiñonero-Candela et al., 2008), where the test distribution, in general, is unknown and different from training distribution: distribution shifts might occur either geographically or temporally (Koh et al., 2021; Shen et al., 2021). A large fraction of research lies in Domain Generalization (DG) (Duchi et al., 2011; Sagawa et al., 2019; Yan et al., 2020; Wang et al., 2020d; Li et al., 2018; Ganin et al., 2016; Arjovsky et al., 2019), aiming to find the shared invariant mechanism across all environments/domains. However, there are no effective DG methods that can outperform Empirical Risk Minimization (ERM) (Vapnik, 1998) on all datasets, as pointed out by Gulrajani & Lopez-Paz (2020). An orthogonal line of works, in contrast to DG methods which require accessing multiple domains for domain invariant learning, focuses on training on a single domain and evaluating the robustness on test datasets with common corruptions (Hendrycks & Dietterich, 2019b) and natural distribution shifts (Recht et al., 2018; 2019; Taori et al., 2020; Hendrycks et al., 2021b;a).

Investigating OOD generalization for FL is a timely topic for making FL realistic. Existing works, merely contemplate optimizing a distributionally robust global model and therefore fall short of achieving impressive test-time accuracy. For example, DRFA (Deng et al., 2020b) adapts Distributionally Robust Optimization (DRO) (Duchi et al., 2011) to FL and optimizes the agnostic (distributionally robust) empirical loss in terms of combining different local losses with learnable weights. FedRobust (Reisizadeh et al., 2020) introduces local minimax robust optimization per client to address the specific affine distribution shift across clients. GMA (Tenison et al., 2022) uses a masking strategy on client updates to learn the invariant mechanism across clients. A similar concept can be found in fairness-aware FL (Mohri et al., 2019; Li et al., 2020c; 2021b;a; Du et al., 2021; Shi et al., 2021), in which performance stability for the global model is enforced across different local training distributions.

Our work improves the robustness of FL deployment under various types of distribution shifts via Test-time Adaptation, and is orthogonal and complimentary to the aforementioned methods.

**Benchmarking FL with distribution shift.** Existing FL benchmarks (Ryffel et al., 2018; Caldas et al., 2018; He et al., 2020a; Yuchen Lin et al., 2022; Lai et al., 2022; Wu et al., 2022b) do not properly measure the test-performance of PFL, or consider various test distribution shifts. Other OOD benchmarks, such as Wilds (Koh et al., 2021; Sagawa et al., 2021) and DomainBed (Gulrajani & Lopez-Paz, 2020), primarily concern the generalization over different domains (with shifts), and hence cannot be directly used to evaluate PFL.

Our proposed benchmark for robust FL (BRFL) in section 5 fills this gap, by covering scenarios specific to FL made up of the co-variate distribution shift and label distribution shift. Note that while the concurrent work (Wu et al., 2022b) similarly argues the sensitivity of PFL to the label distribution shift, they—unlike this work—neither consider other complex and realistic scenarios, nor offer a solution.

**Test-Time Adaptation (TTA)** was initially presented in Sun et al. (2020) and has garnered growing interest: it improves the accuracy for arbitrary pre-trained models, including those trained using robustness methods. Test-Time Training (TTT) (Sun et al., 2020) relies on a two-head neural network structure, in which the feature extractor is fine-tuned via a head with a self-supervised task, improving the accuracy for the prediction branch. TTT++ (Liu et al., 2021) adds a feature alignment block to (Sun et al., 2020), but it also necessitates accessing the entire test dataset prior to testing, which may not be possible during online deployment. Tent (Wang et al., 2020a) minimizes the prediction entropy and only updates the Batch Normalization (BN) (Ioffe & Szegedy, 2015) parameters, while MEMO (Zhang et al., 2021a) adapts all model parameters through minimizing

the entropy of marginal distribution over a set of data augmentations. TSA (Zhang et al., 2021c) improves long-tailed recognition task via interpolating skill-diverse expert heads predictions, where head aggregation weights are optimized to handle unknown test class distribution. T3A (Iwasawa & Matsuo, 2021) replaces a trained linear classifier by class prototypes and classifies each sample based on its distance to prototypes. CoTTA (Wang et al., 2022) instead considers a continual scenario, where class-balanced test samples will encounter different corruption types sequentially.

Compared to the existing TTA methods designed for non-distributed and homogeneous cases, our two-head approach is uniquely motivated by FL scenarios: we are the first to build effective yet efficient online TTA methods for PFL on heterogeneous training/test local data with FL-specific shifts. Our solution is intrinsically different from TTT variants—rather than using the auxiliary task head to optimize the feature extractor for the prediction task head, our design focuses on two FL-specific heads learned with global and local objectives and learns the optimal head interpolation weight.

## B    DETAILED ALGORITHMS

**Training phase of FedTHE and FedTHE+.**    In Algorithm 1, we illustrate the detailed training procedure of our scheme. The scheme is consistent with training algorithm such as FedAvg (McMahan et al., 2017) For each communication round, the client performs local training by training the global model received from server and tuning personalized head on top of the received and frozen global feature extractor. The parameters from local training (i.e. $\mathbf{w}^e$ and $\mathbf{w}^g$) along with the local descriptor $\mathbf{h}^l$ are sent to the server for aggregation. The server averages the local descriptor $\mathbf{h}^l$ from clients to generate the global descriptor $\mathbf{h}^g$ and executes a model aggregation step using received local model parameters. Note that these procedures is applicable to both FedTHE and FedTHE+.

---

**Algorithm 1** FedTHE/ FedTHE+ (federated training phase).

---

**Require:** Number of clients $K$; client participation ratio $r$; step size $\eta$; number of local and personalized training updates $\tau$ and $p$; number of communication rounds $T$; initial feature extractor parameter $\mathbf{w}^e$ and global head parameter $\mathbf{w}^g$; initial personalized head parameters $\mathbf{w}^{l_1}, \ldots, \mathbf{w}^{l_n}$.

1: **for** $t \in \{1, \ldots, T\}$ **do**
2:     server samples a set of clients $\mathcal{S} \subseteq \{1, \ldots, K\}$ with size $rK$
3:     server sends $\mathbf{w}^e, \mathbf{w}^g$ to the selected clients $\mathcal{S}$
4:     **for each** sampled client $m$ **do**
5:        client $m$ initialize $\tilde{\mathbf{w}}_m^e := \mathbf{w}^e, \tilde{\mathbf{w}}_m^g := \mathbf{w}^g, \tilde{\mathbf{w}}_m^l := \mathbf{w}^l$
6:        **for** $s \in \{1, \ldots, \tau\}$ **do**
7:           $\tilde{\mathbf{w}}_m^e, \tilde{\mathbf{w}}_m^g = \text{MiniBatchSGDUpdate}(\tilde{\mathbf{w}}_m^e, \tilde{\mathbf{w}}_m^g; \eta)$     $\triangleright$ train extractor & global head
8:        **for** $s \in \{1, \ldots, p\}$ **do**
9:           $\tilde{\mathbf{w}}_m^l = \text{MiniBatchSGDUpdate}(\mathbf{w}_m^e, \tilde{\mathbf{w}}_m^l; \eta)$        $\triangleright$ train personalized head
10:        client $m$ collects local feature $\tilde{\mathbf{h}}_m^l$ and sends $\tilde{\mathbf{w}}_m^e, \tilde{\mathbf{w}}_m^g, \tilde{\mathbf{h}}_m^l$ to server
11:     $\mathbf{w}^e = \sum_{m \in \mathcal{S}} \frac{|\mathcal{D}_m|}{\sum_{l \in \mathcal{D}} |\mathcal{D}_l|} \tilde{\mathbf{w}}_m^e$        $\triangleright$ server aggregates the parameters of feature extractor
12:     $\mathbf{w}^g = \sum_{m \in \mathcal{S}} \frac{|\mathcal{D}_m|}{\sum_{l \in \mathcal{D}} |\mathcal{D}_l|} \tilde{\mathbf{w}}_m^g$        $\triangleright$ server aggregates the parameters of global head
13:     $\mathbf{h}^g = \sum_{m \in \mathcal{S}} \frac{|\mathcal{D}_m|}{\sum_{l \in \mathcal{D}} |\mathcal{D}_l|} \tilde{\mathbf{h}}_m^l$        $\triangleright$ server aggregates the local features
14: **return** $\mathbf{w}^e, \mathbf{w}^g$ and $\mathbf{h}^g$

---

**Test-time robust online deployment using FedTHE & FedTHE+.**    In Algorithm 2, we demonstrate the detailed test-time online deployment phase of FedTHE+ (FedTHE can be seen as the Linear-Probing phase, from line 1 to 9), which only requires a single unlabeled test sample to perform test-time robust personalization. Specifically, when a new test-sample $\mathbf{x}_n$ arrives,

1. we first do forward pass to obtain a test feature $\mathbf{h}_n$, and use test history descriptor $\mathbf{h}^{\text{history}}$ to stabilize this feature. After that, several optimization steps are performed on head ensemble weight $e$, using the adaptive unsupervised loss constructed by Feature Alignment (FA), Entropy Minimization (EM) and Similarity-based Loss Weighting (SLW). Once the optimization ends, the test history is updated using the original (i.e. with no stabilization) test feature $\mathbf{h}_n$.

FedTHE is essentially the aforementioned phase, which may be viewed as a way of performing unsupervised Linear-Probing.

2. FedTHE+ is a slight extension of FedTHE, where a test-time fine-tuning method (e.g. MEMO (Zhang et al., 2021a) in Algorithm 3) is performed on top of FedTHE: the additional procedure of FedTHE+ mimics the Fine-Tuning phase of LP-FT.

---

**Algorithm 2** FedTHE+ (test-time online deployment per test sample).

---

**Require:** Feature extractor parameter $\mathbf{w}^e$; global head parameter $\mathbf{w}^g$; local head parameter $\mathbf{w}^l$; test history $\mathbf{h}^{\text{history}}$, global feature $\mathbf{h}^g$ and local feature $\mathbf{h}^l$; a single test sample $\mathbf{x}_n$; ensemble weight optimization steps $s$; test-time FT steps $\tau$; initial learnable head ensemble weight $e = 0.5$; $\alpha = 0.1$; $\beta = 0.3$; step size $\eta$.

1: $\mathbf{h}_n = f_{\mathbf{w}^e}(\mathbf{x}_n)$          ▷ do forward pass to get a test feature for unknown $\mathbf{x}_n$
2: $\mathbf{h}'_n = \beta\mathbf{h}_n + (1 - \beta)\mathbf{h}^{\text{history}}$          ▷ stabilize the test feature via test history feature
3: $\hat{\mathbf{y}}^g, \hat{\mathbf{y}}^l = f_{\mathbf{w}^g}(\mathbf{h}_n), f_{\mathbf{w}^l}(\mathbf{h}_n)$          ▷ get global & local head logit outputs
4: initialize test-time deployment: $\tilde{\mathbf{w}}^e, \tilde{\mathbf{w}}^g, \tilde{\mathbf{w}}^l = \mathbf{w}^e, \mathbf{w}^g, \mathbf{w}^l$
5: **for** $t \in \{1, \dots, s\}$ **do**
6:     $e = e - \eta\nabla_e\mathcal{L}_{\text{THE}}(e; \hat{\mathbf{y}}^g, \hat{\mathbf{y}}^l, \mathbf{h}^g, \mathbf{h}^l, \mathbf{h}'_n)$      ▷ optimize head ensemble weight based on (5)
7: $e^\star = e$
8: $\mathbf{h}^{\text{history}} = \alpha\mathbf{h}_n + (1 - \alpha)\mathbf{h}^{\text{history}}$
9: **for** $t \in \{1, \dots, \tau\}$ **do**
10:     $\tilde{\mathbf{w}}^e, \tilde{\mathbf{w}}^g, \tilde{\mathbf{w}}^l = \text{MiniBatchSGD}_{\text{FT}}(\tilde{\mathbf{w}}^e, \tilde{\mathbf{w}}^g, \tilde{\mathbf{w}}^l; \mathbf{x}_n, e^\star, \eta)$      ▷ test-time FT (e.g. MEMO)
11: $\mathbf{y}^\star = f_{e^\star\tilde{\mathbf{w}}^g + (1-e^\star)\tilde{\mathbf{w}}^l}\left(f_{\tilde{\mathbf{w}}^e}(\mathbf{x}_n)\right)$
12: **return** final logits prediction $\mathbf{y}^\star$

---

**Algorithm 3** MEMO (Zhang et al., 2021a).

---

**Require:** A trained model $f_{\mathbf{w}}$, including feature extractor and prediction head; a single test sample $\mathbf{x}$; step size $\eta$; # of augmentations $B$; a complete set of augmentation functions $\mathcal{A} \triangleq \{a_1, \dots, a_M\}$; test-time FT steps $\tau$

1: sample $a_1, \dots, a_B \stackrel{\text{i.i.d.}}{\sim} \mathcal{U}(\mathcal{A})$ and produce augmented points $\tilde{\mathbf{x}}_i = a_i(\mathbf{x})$ for $i \in \{1, \dots, B\}$
2: **for** $t \in \{1, \dots, \tau\}$ **do**
3:     Monte Carlo estimation of marginal output distribution $\hat{\mathbf{y}} := \frac{1}{B}\sum_{i=1}^B f_{\mathbf{w}}(\tilde{\mathbf{x}}_i)$
4:     compute entropy loss $\mathcal{L} = \text{entropy}(\hat{\mathbf{y}}) = -\sum_i p_i(\hat{\mathbf{y}}) \log p_i(\hat{\mathbf{y}})$
5:     adapt model parameters $\mathbf{w} = \mathbf{w} - \eta\nabla_{\mathbf{w}}\mathcal{L}$      ▷ arbitrary optimizer
6: **return** $f_{\mathbf{w}}(\mathbf{x})$

---

## C  DETAILED CONFIGURATION AND EXPERIMENTAL SETUPS

### C.1  TRAINING, DATASETS, AND MODEL CONFIGURATIONS FOR (PERSONALIZED) FL.

**Training setups.** For the sake of simplicity, we consider a total of 20 clients with a participation ratio of 1.0 for the (personalized) FL training process, and train for 100 and 300 communication rounds for CIFAR10 and ImageNet-32 (i.e. image resolution of 32), respectively. We decouple the local training and personalization phases for every communication round to better understand their impact on personalized FL performance. More precisely, a local 5-epoch training is performed on the received global model; the local personalization phase again considers the same received global model, and we use 1 epoch in our cases. Only the parameters from local training phase will be sent to server and aggregated for global model. We also investigate the impact of different local personalized training epochs in Appendix E.4 and find that 1 personalization epoch is a good trade-off point for time complexity and performance.

For all the experiments, we train CIFAR10 with a batch size of 32 and ImageNet-32 with that of 128. Results in all figures and tables are reported over 3 different random seeds. We elaborate the detailed configurations for different methods/phases below:

- For head ensemble phase (personalization phase) of FedTHE and FedTHE+, we optimize the head ensemble weight $e$ by using a Adam optimizer with initial learning rate 0.1 (when training CNN or ResNet20-GN) or 0.01 (when training CCT4), and 20 optimization steps. The head ensemble weight $e$ is always initialized as 0.5 for each test sample and we use $\alpha = 0.1$ and $\beta = 0.3$ for feature alignment phase. These configurations are kept constant throughout all the experiments (different neural architectures, datasets).

- We further list configurations and hyperparameters used for local training and local personalization. Note that our methods FedTHE and FedTHE+ also rely on them to train feature extractor, global head, and personalized head.

  For training CNN on CIFAR10, we use SGD optimizer with initial learning rate 0.01. We set weight decay to 5e-4, except for Ditto we use zero weight decay as it has already included regularization constraints.

  For training ResNet20-GN (He et al., 2016) on both CIFAR10 and ImageNet-32, we use similar settings as training CNN on CIFAR10, except we use SGDm optimizer with momentum factor of 0.9. We set the number of group to 2 for GroupNorm(Wu & He, 2018) in ResNet20-GN, as suggested in Hsieh et al. (2020).

  For training CCT4 (Hassani et al., 2021) on CIFAR10, we use Adam (Kingma & Ba, 2014) optimizer with initial learning rate 0.001 and default coefficients (i.e. 0.9, 0.999 for first and second moments respectively).

**Remarks on the number of clients.**   Our experimental setup is extended from FedRod (Chen & Chao, 2022), which also considers 20 clients when conducting experiments on CIFAR10/100. Similar settings can also be found in some famous FL benchmarks, e.g., He et al. (2020a). Also, our setting is better aligned and of higher importance to the cross-silo FL settings, where 20 clients is a reasonable system scale and robustness is crucial. Finally, splitting the datasets to a lot more clients would suffer from lacking data issues. For example, CIFAR10.1 (the only existing natural co-variate shift dataset for CIFAR) only contains 2000 samples, splitting it into 100 or more clients will result in less than 20 natural co-variate shifted test samples for each client. As the first step in considering test-time robust FL deployment, we look forward to more suitable datasets from the community to support more client partition. We also add results of 100 clients in subsection E.4 in case of concerns on number of clients.

**Models.**   We use the last fully connected (FC) layer as generic global head and personalized local head for CNN, ResNet20-GN, and CCT4. Similar to (Chen & Chao, 2022), we investigate the effect of using different numbers of FC layers as heads; however, more FC layers only give sub-optimal results. The neural architecture of CNN contains 2 Conv layers (with 32, 64 channels and 5 kernel size) ans 2 FC layers (with 64 neurons in the hidden size). For ResNet20, we set the scaling factor of width to 1 and 4 for training on CIFAR10 and ImageNet-32, respectively. For CCT4 on CIFAR10, we use the exact same architecture as in Hassani et al. (2021) for CCT-4/3x2 with a learnable positional embedding. The dimension of feature representation (i.e. the output dimension of feature extractor) of CNN and ResNet20 on CIFAR10, and ResNet20 on ImageNet-32 and CCT4 are 64, 64, 256, 128, respectively. The superior performance of our methods on different architecture indicates that our approaches are not affected by the dimension of feature representations.

**Datasets.**   For CIFAR10, we follow the standard preprocessing procedure, where we augment the training set through horizontal flipping and random crop, and normalize the input to $[-1, 1]$ for both training and test sets (i.e. using (0.5,0.5,0.5) for mean and std).

For ImageNet-32, we apply the same augment process as CIFAR10. Besides, we resize its corresponding OOD datasets (i.e. ImageNet-A (Hendrycks et al., 2021b), ImageNet-R (Hendrycks et al., 2021a), and ImageNet-V2 (Recht et al., 2019)) to $32 \times 32$, where bi-linear interpolation is used.

## C.2 HYPERPARAMETERS TUNING

We tune the hyperparameters for each baseline. We first tune the hyperparameters for each baseline on Dir(0.1) heterogeneous local distribution, and then further tune the hyperparameters (in a narrower range for computational feasibility starting from optimal hyperparameters on Dir(0.1)) for the Dir(1.0) case. We can witness that most of existing methods are not sensitive to the degree of data heterogeneity.

It is worth emphasizing that obtaining a proper and fair hyperparameter tuning principle for test-time scenarios (i.e. deployment time) is still an open question, and we leave this for future work.

- For GMA (Tenison et al., 2022), we tune the masking threshold in $\{0.1, 0.2, 0.3, ..., 1.0\}$ and observe that 0.1 is the best one.
- For FedRep (Collins et al., 2021) and FedRoD (Chen & Chao, 2022), we use the last FC layer as the head, as what FedRep did in their original paper.
- For APFL (Deng et al., 2020a), we use the adaptive $\alpha$ scheme, where the interpolation $\alpha$ of clients are updated per-round, and the update step of $\alpha$ follows the setting in subsection C.1 (following the treatment in the original paper).
- For Ditto (Li et al., 2021b), we tune the regularization factor $\lambda$ in the grids of $\{0.01, 0.1, 1.0, 5.0\}$ and 1.0 is the final choice.
- For kNN-Per (Marfoq et al., 2022), we follow their setting: the number of neighbors $k$ is set to 10 and the scaling factor $\sigma$ is set to 1.0. For the more crucial hyperparameter $\lambda_m$, we do a more fine-grained grid search with a 0.01 interval (rather than a 0.1 interval in their experiments) in $\{0.0, 0.01, 0.02, ..., 0.99 1.0\}$ for each client, to ensure the optimal values are used for in-distribution data.
- For MEMO (Zhang et al., 2021a), we select the number of augmentations from $\{16, 32, 48\}$ and number of optmization steps from $\{1, 2, 3, 4\}$. We find 32 augmentations and 3 optimization steps with learning rate 0.0005 reaches the best performance.
- For T3A (Iwasawa & Matsuo, 2021), we select the size of support set $M$ from $\{1, 5, 20, 50, 100, N/A\}$ (where $N/A$ means do not filter the support set) and find $M = 50$ is the best choice.
- For TTT (Sun et al., 2020), we tune the learning rate from $\{0.0001, 0.001, 0.01\}$ and optimization steps from $\{1, 2, 3\}$, and find 0.001 learning rate and 1 optimization step is the optimal.
- For FedTHE (ours), we use $\alpha = 0.1$ and $\beta = 0.3$ throughout all the experiments. We provide additional ablation study in Appendix E.4 for the hyperparameters $\alpha$ in the grids of $\{0.05, 0.1, 0.2\}$ and $\beta$ in the grids of $\{0.1, 0.2, 0.3, 0.4, 0.5\}$.
- For FedTHE+ (ours), we further tune the test-time fine-tuning phase (MEMO in our case), and choose to use 16 augmentations and 3 optimization steps for our MEMO, with 0.0005 learning rate. No learning rate decay is added during test-time fine-tuning phase. All these hyperparameters are kept constant throughout all the experiments.

## D ADDITIONAL DETAILS FOR OUR BENCHMARK FOR ROBUST FL (BRFL)

Throughout the experiments, we construct various test distributions for assessing the robustness of (personalized) FL models during deployment. We provide more details here.

As shown in Figure 4, taking CIFAR-10 as an example, Dirichlet distribution is used to partition the training set into disjoint non-i.i.d local client datasets and each client's dataset is further split into local training/validation/test set, where the local training/validation set are used to train/tune hyperparameters and the local test set is formulated to different test cases to evaluate test-time robustness. We construct 5 distinct test distributions, namely ⓐ original local test, ⓑ synthetic corrupted local test, ⓒ naturally shifted test, ⓓ original Out-of-Client local test and ⓔ mixture of test. In details:

ⓐ is supposed to reflect how well the model adapts to local training distribution. Specifically, The training set is partitioned to disjoint local subsets by Dirichlet distribution, and each local subset is further split to disjoint local training set and Original local test set.

ⓑ mimics the common corruptions (synthetic distribution shift) on local test distribution. For each test sample from ⓐ, we randomly sample a corruption from the 15 common corruptions (Hendrycks & Dietterich, 2019a) and apply it to the test sample. We set the severity level of corruptions to highest 5 for all experiments, and we also investigate different severity in Appendix E.4.

ⓒ mimics the real-world natural distribution shift. We record the class distribution of each client, and split CIFAR10.1 into subsets where each subset's class distribution is consistent with the corresponding client's local training set class distribution.

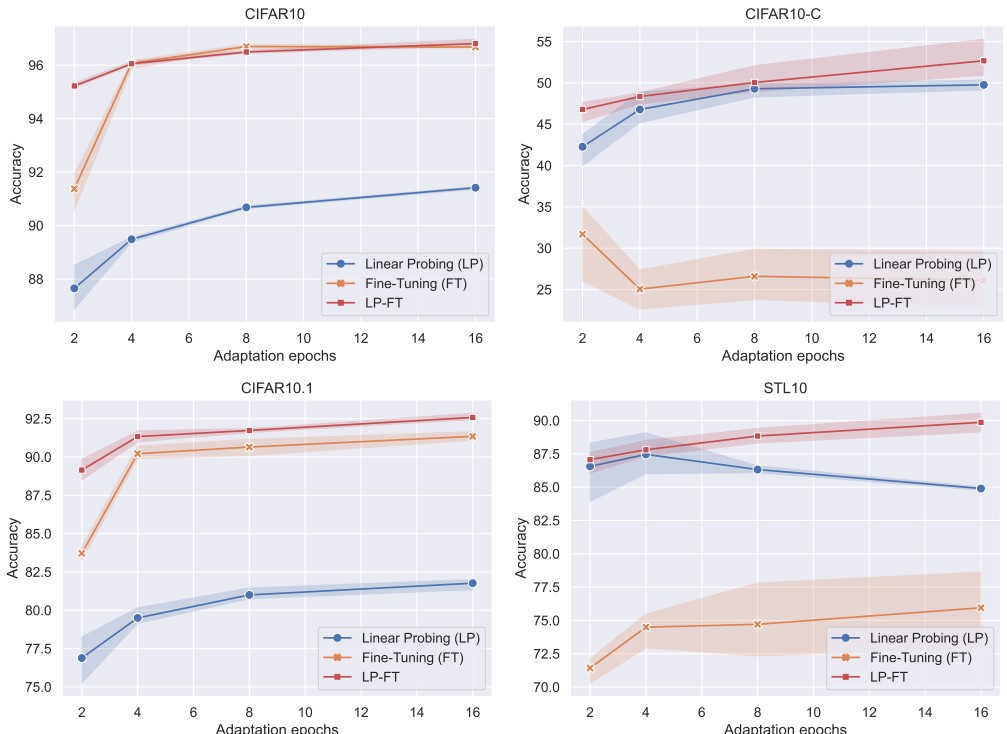

Figure 6: **The test performance of using different adaptation strategies** (i.e. LP, FT and LP-FT) on **MoCoV2** pre-trained ResNet50. 1 ID data (CIFAR10) and 3 OOD data (CIFAR10-C, CIFAR10.1 and STL10) are considered.

ⓓ mimics the label distribution shift. We randomly sample other clients' test samples to form a test set of the same size of original local test set, and because the existence of non-i.i.d, such sampling can be seen as a case of label distribution shift.

ⓔ is constructed by mixing the test samples in random order from the previous 4 cases.

For downsampled ImageNet-32, we leverage the same process to build ⓔ, and we use the same procedure as ⓓ to construct 3 naturally shifted test sets from ImageNet-A, ImageNet-R, ImageNet-V2, and we use the same procedure as ⓔ to build mixture of test. Note that ImageNet-A and ImageNet-R only contains a subset of 200 classes of the original 1000 classes of ImageNet, and only a small portion of the two subsets overlap. To remedy this, we take the intersection of ImageNet-A and R classes which contains 86 classes and then we filter the downsampled ImageNet-32 and ImagNet-V2 to have the same class list.

For both CIFAR-10 and downsampled ImageNet-32, after obtaining the performance on 5 test distributions (1 ID data and 4 OOD data), we simply compute their average performance and standard deviation as a more straightforward metric, as shown in subsection 6.2.

# E  ADDITIONAL RESULTS AND ANALYSIS

## E.1  JUSTIFICATION OF THE EFFECTIVENESS OF LP-FT

Evidences on the intuition of designing FedTHE based on LP-FT scheme are shown here. In Figure 6 and Figure 7, we perform LP, FT, and LP-FT on MoCoV2 and its weaker version MoCo V1 (He et al., 2020b; Chen et al., 2020b) on pretrained ResNet-50 (He et al., 2016), on both ID & OOD data of CIFAR10: ● CIFAR10 original test set as ID data; ● CIFAR10-C (Hendrycks & Dietterich, 2018) and CIFAR10.1 (Recht et al., 2018) as OOD data; ● additionally, we add the STL-10 (Coates et al., 2011) (which is a standard domain adaptation dataset) as one more OOD data, similar to (Kumar

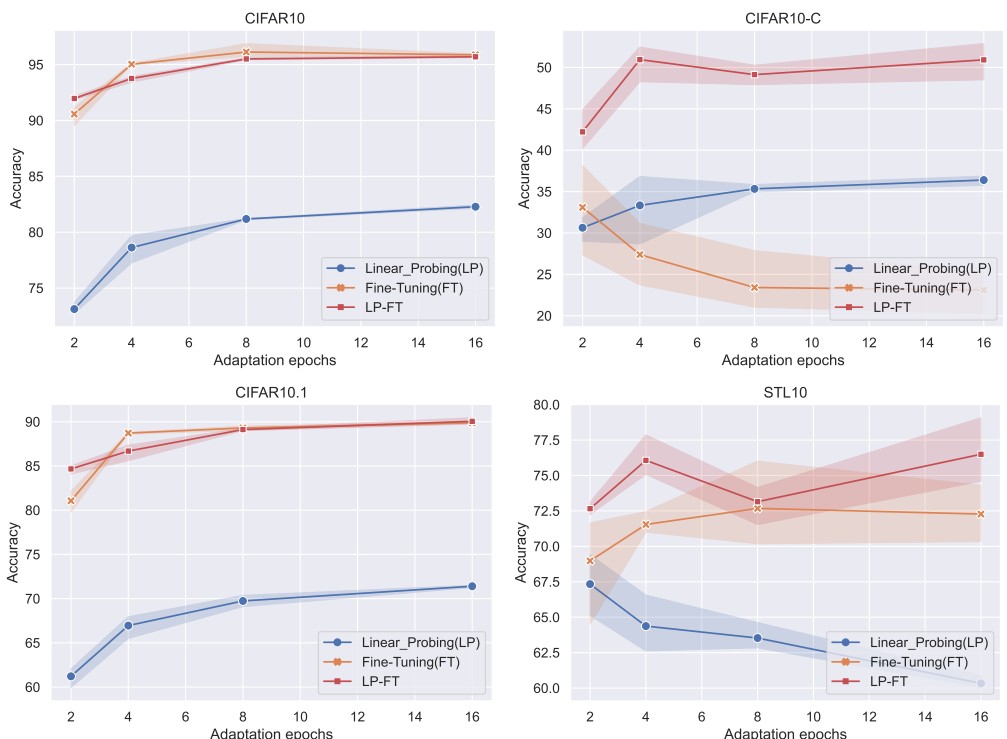

Figure 7: **The test performance of using different adaptation strategies** (i.e. LP, FT and LP-FT) on **MoCoV1** pre-trained ResNet50. 1 ID data (CIFAR10) and 3 OOD data (CIFAR10-C, CIFAR10.1 and STL10) are considered.

et al., 2022)[10]. The same conclusion as we draw in section 3 is observed: while maintaining a better OOD performance, LP-FT does not sacrifice ID performance.

### E.2    DETAILED ANALYSIS FOR TRAINING CNN ON CIFAR10

We provide more details for Table 1, in terms of learning curve, for better illustration. In Figure 8 and Figure 9, we show a more comprehensive comparison of the learning curve of mixture of test accuracy for different baselines. Our method FedTHE+ significantly outperforms all the other baselines on this mixture of test distributions, indicating our method's robustness in real-world deployment scenarios. In Figure 10 and Figure 11, we show the advancement of our method on all test cases compared to the strong baselines from FL, Personalized FL and Test-time adaptation. Also, one can see the significant performance drop of different test distribution shift cases compared to the original local test distribution, which demonstrates the importance of robustifying FL models under ID and OOD data distribution during deployment.

To demonstrate the advancement of our method on even longer federated learning, in Figure 12 and Figure 13, we show the extended 500 and 1000 communication rounds learning curve of representative methods on the mixture of test. The results indicate that our method always outperforms the other methods by a large margin, and the conclusion that our method accelerates the federated optimization by 1.5x-2.5x still hold.

### E.3    MORE RESULTS FOR DIFFERENT DEGREES OF NON-I.I.D AND A MORE CHALLENGING
         TEST CASE

For the purpose of showing FedTHE and FedTHE+ generalizes on broader choices of the non-i.i.d degree of local data distribution, including on different datasets (i.e. CIFAR10 and downsampled ImageNet-32) and different architectures (i.e. ResNet20-GN and CCT4), we provide more results and detailed numerical results. We also create a more challenging test case by adding two types

---

[10]We take the overlapping 9 class of CIFAR10 and STL-10 for evaluating performance of LP, FT and LP-FT.

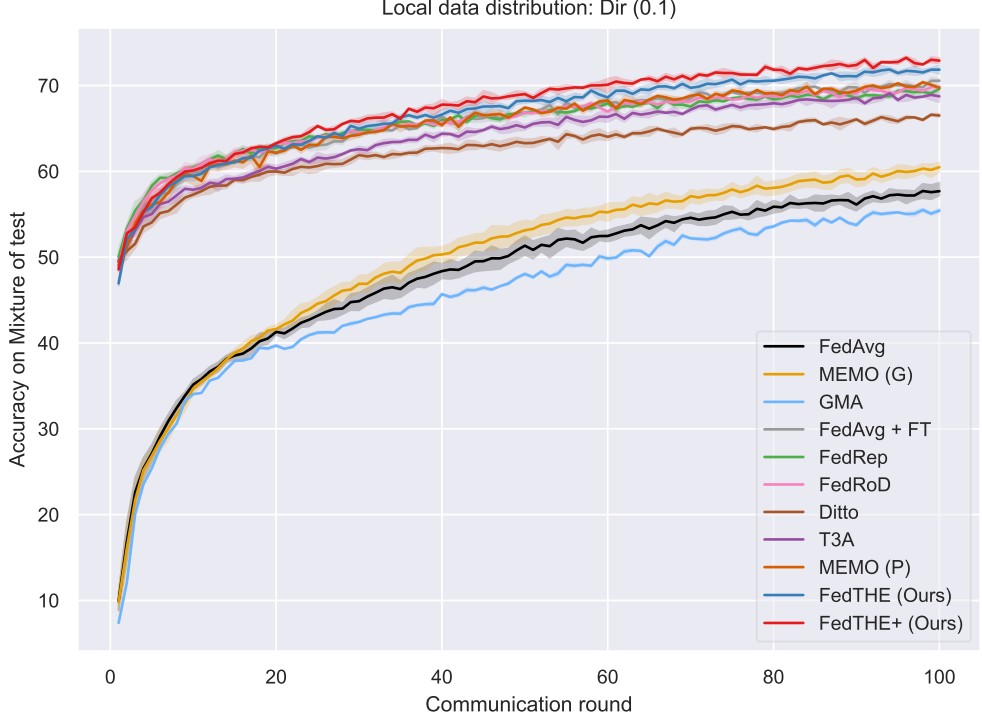

Figure 8: **The learning curves (train CNN on CIFAR10) for different baselines** (including FL, personalized FL and test-time adaptation methods) on **mixture of test** and **Dir(0.1)** non-i.i.d local distributions.

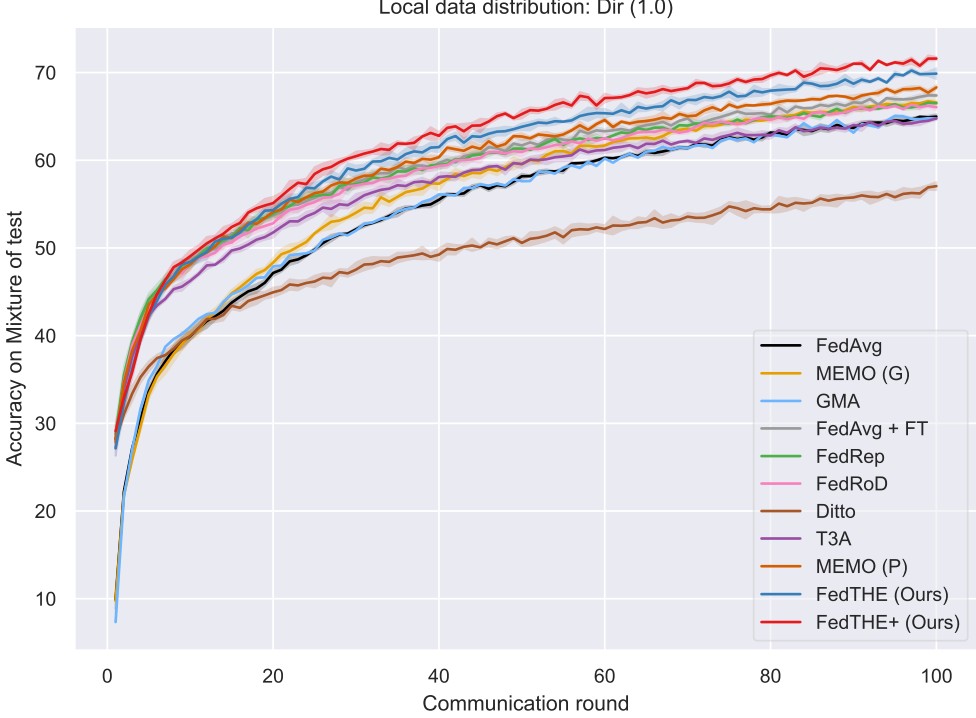

Figure 9: **The learning curves (train CNN on CIFAR10) for different baselines** (including FL, personalized FL and test-time adaptation methods) on **mixture of test** and **Dir(1.0)** non-i.i.d local distributions.

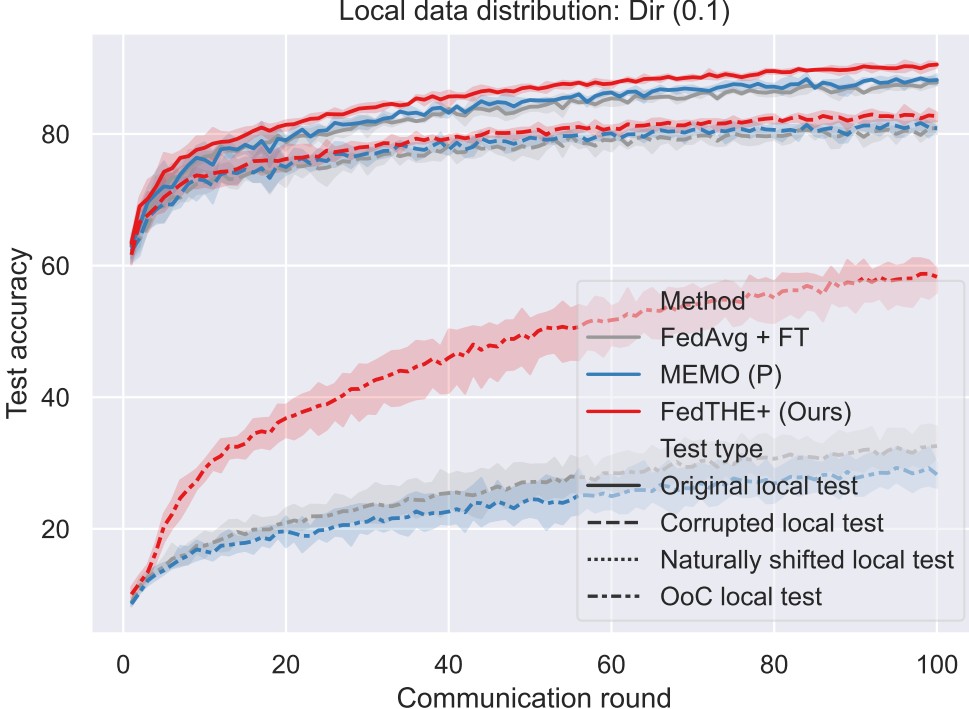

Figure 10: **The learning curves (train CNN on CIFAR10) for strong baselines** (including FL, personalized FL and test-time adaptation methods) on **original local test, corrupted local test, naturally shifted local test and original out-of-client local test and Dir(0.1)** non-i.i.d local distributions.

of distribution shifts on each test sample and demonstrate the advanced robustness of FedTHE and FedTHE+ in this case.

**Different degrees of non-i.i.d local distributions.**   In Table 4, we use our benchmark to investigate the results of training CNN on CIFAR10 on Dir(10.0). While most of the baseline fail to outperform FedAvg in this weak non-i.i.d scenario, our method still gives noticeable results and also on the more challenging mixture of test case, indicating that our method is powerful on a broad range of non-i.i.d degree. In Table 5, we also provide results for training ResNet20-GN on downsampled ImageNet-32 on Dir(0.1). Similar to the Dir(1.0) case (Table 2 in subsection 6.2), significant performance gain is achieved on almost all test cases. We hypothesize that this is because the fast adaptation nature of our first phase FedTHE as Linear-Probing. In Table 6, we show the numerical results of Figure 5 (in the main paper) for more clear illustration. We see that more significant performance gain is achieved when switching to more powerful architectures.

**Evaluating on interleaving test distribution shifts.**   In Table 7, we create a more challenging test case where each test sample contains two distribution shifts (i.e. adding synthetic corruption on original Out-of-Client test set). By evaluating the performance of representative personalized FL methods on different degrees of non-i.i.d local distributions and architectures, we see that both FedTHE and FedTHE+ outstandingly outperform the other baselines for around %10 at least. This indicates that our method is also robust on interleaving test distribution shifts, making test-time robust deployment a lot easier. We leave the other interleaving cases of test distributions for future work.

### E.4   MORE ABLATION STUDY FOR FEDTHE AND FEDTHE+

**Ablation study for training epochs, dataset sizes and corruption severity.**   Here we provide more details about the omitted ablation study due to the space constraint. In Table 8–Table 11, we compare MEMO (P), which is a strong baseline, with FedTHE+ in terms of local personalized training epochs (i.e., training on personalized head), train dataset size and corruption severity. The results indicate

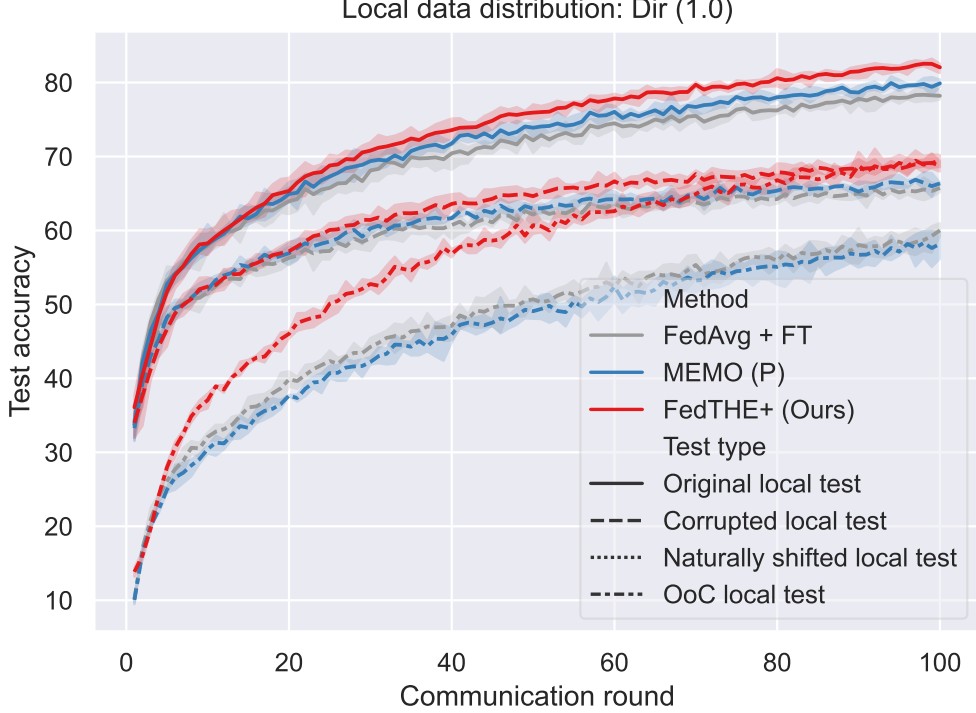

Figure 11: **The learning curves (train CNN on CIFAR10) for strong baselines** (including FL, personalized FL and test-time adaptation methods) on **original local test, corrupted local test, naturally shifted local test and original out-of-client local test and Dir(1.0)** non-i.i.d local distributions.

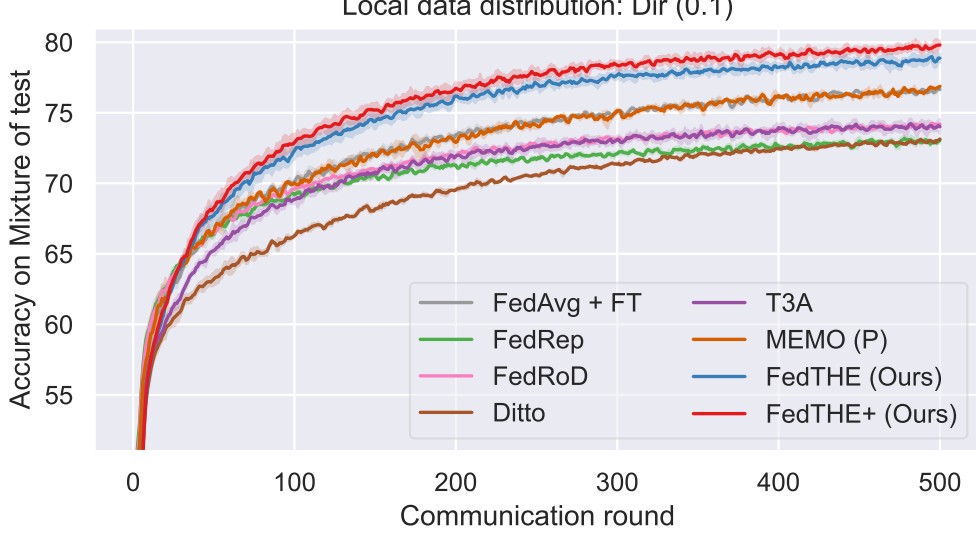

Figure 12: **The learning curves (train CNN on CIFAR10) for strong baselines** (including FL, personalized FL and test-time adaptation methods) on **mixture of test and Dir(0.1)** non-i.i.d local distributions.

that FedTHE+ still outperforms MEMO (P) regardless the choice of training and dataset settings, meaning our method is insensitive to different FL training settings and datasets, etc. Also, in Table 10, we conduct experiments regarding different sizes of test set, because both FedTHE and FedTHE+ require maintaining a test history, which depends on the number of test samples seen. As we can see, there are marginal differences when we vary the size of local test set, meaning that our methods are not sensitive to local test set size.

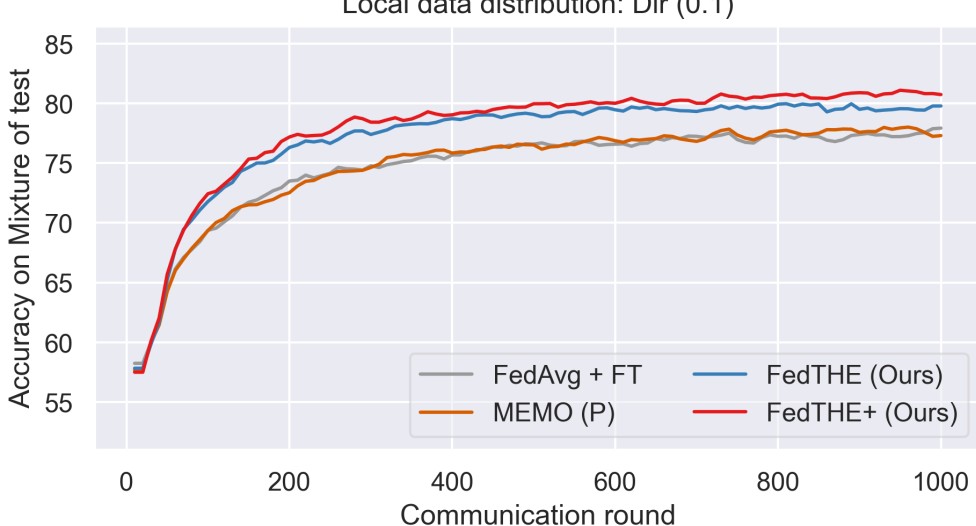

Figure 13: **The learning curves (train CNN on CIFAR10) for 1000 rounds on strong baselines** (including FL, personalized FL and test-time adaptation methods) on **mixture of test and Dir(0.1)** non-i.i.d local distributions.

Table 4: **Test top-1 accuracy of (personalized) FL methods across various test distributions**, for learning on clients with **Dir(10.0).** The experiments are evaluated on a simple CNN architecture for CIFAR10. The adaptation hyperparameters are tuned for each baseline, and the results are averaged over three random seeds. Our approaches' hyperparameters are kept constant throughout the experiments.

| Methods | Local data distribution: Dir(10.0) | | | | |
|---|---|---|---|---|---|
| | Original local test | Corrupted local test | Original OoC local test | Naturally Shifted test | Mixture of test |
| FedAvg | $76.28_{\pm0.77}$ | $59.51_{\pm0.67}$ | $76.27_{\pm0.27}$ | $60.63_{\pm0.93}$ | $68.18_{\pm0.58}$ |
| GMA | $72.62_{\pm0.60}$ | $57.51_{\pm0.76}$ | $72.52_{\pm0.27}$ | $56.92_{\pm1.32}$ | $64.89_{\pm0.64}$ |
| MEMO (G) | $78.19_{\pm0.40}$ | $60.09_{\pm0.46}$ | $\mathbf{78.36}_{\pm0.29}$ | $\mathbf{63.31}_{\pm0.28}$ | $69.98_{\pm0.22}$ |
| FedAvg + FT | $74.55_{\pm1.00}$ | $57.68_{\pm0.59}$ | $71.98_{\pm0.33}$ | $59.15_{\pm0.94}$ | $65.83_{\pm0.67}$ |
| APFL | $56.12_{\pm0.96}$ | $46.84_{\pm0.54}$ | $45.17_{\pm0.98}$ | $39.71_{\pm0.28}$ | $46.96_{\pm0.47}$ |
| FedRep | $75.24_{\pm0.24}$ | $58.05_{\pm1.04}$ | $70.81_{\pm0.61}$ | $58.16_{\pm0.94}$ | $65.56_{\pm0.30}$ |
| Ditto | $62.46_{\pm1.94}$ | $50.80_{\pm0.64}$ | $50.81_{\pm0.58}$ | $42.71_{\pm0.45}$ | $51.69_{\pm0.85}$ |
| FedRoD | $73.96_{\pm0.27}$ | $57.49_{\pm0.68}$ | $70.58_{\pm0.32}$ | $57.88_{\pm0.95}$ | $64.97_{\pm0.22}$ |
| TTT | $74.56_{\pm2.02}$ | $57.43_{\pm1.04}$ | $70.64_{\pm0.47}$ | $59.15_{\pm1.63}$ | $65.44_{\pm1.23}$ |
| MEMO (P) | $76.54_{\pm0.46}$ | $58.91_{\pm0.25}$ | $73.97_{\pm0.64}$ | $61.59_{\pm0.41}$ | $67.76_{\pm0.15}$ |
| T3A | $70.35_{\pm1.10}$ | $52.87_{\pm0.75}$ | $68.60_{\pm0.67}$ | $57.91_{\pm0.86}$ | $62.42_{\pm0.85}$ |
| FedTHE (Ours) | $77.17_{\pm0.14}$ | $60.47_{\pm0.98}$ | $74.64_{\pm1.12}$ | $59.35_{\pm0.60}$ | $68.84_{\pm0.43}$ |
| FedTHE+ (Ours) | $\mathbf{79.10}_{\pm0.11}$ | $\mathbf{62.34}_{\pm0.62}$ | $76.82_{\pm0.06}$ | $61.48_{\pm0.79}$ | $\mathbf{70.71}_{\pm0.32}$ |

**Ablation study for classifier-level interpolation strategies.** In Figure 14 and Figure 15, we verify the effectiveness of our method through interpolating global and local model. In Figure 14 we use a Y-structure model with shared feature extractor here and we only interpolate the global and local head. While the curve shows the average performance of using a globally shared ensemble weight across clients, its upper bound is achieved by selecting an optimal ensemble weight for each client and average their optimal performance on the mixture of test. Our degraded version FedTHE reaches very close results to upper bound in the two degrees of non-i.i.d local distributions, and FedTHE+ outperforms the upper bound on Dir(0.1) and Dir(1.0). In Figure 15, we investigate the effect of different local/global interpolation strategies, as an additional comparison with other prior personalized FL methods (Deng et al., 2020a; Mansour et al., 2020). *FedTHE significantly outperforms all these strategies*, indicating the effectiveness of our test-time ensemble weight tuning.

Table 5: **Test top-1 accuracy of (personalized) FL methods across different downsampled ImageNet-32 test distributions.** ResNet20-GN with width factor of 4 is used for training ImageNet on clients with heterogeneity **Dir(0.1)**. We compare our methods with five strong competitors on 1 ID data and 4 OODs data (i.e. co-variate natural shifts). We also report the averaged performance over these 5 test scenarios.

| Methods | Local test of ImageNet | ImageNet-A | ImageNet-V2 | ImageNet-R | Mixture of test | Average |
|---|---|---|---|---|---|---|
| FedAvg + FT | $77.82_{\pm 0.99}$ | $16.45_{\pm 0.20}$ | $30.95_{\pm 1.04}$ | $64.47_{\pm 1.55}$ | $47.42_{\pm 0.63}$ | $47.42_{\pm 24.7}$ |
| FedRoD | $78.99_{\pm 0.26}$ | $16.76_{\pm 0.74}$ | $29.83_{\pm 0.49}$ | $63.58_{\pm 0.63}$ | $47.27_{\pm 0.28}$ | $47.27_{\pm 25.0}$ |
| FedRep | $66.07_{\pm 0.68}$ | $15.33_{\pm 0.63}$ | $27.37_{\pm 1.09}$ | $56.65_{\pm 0.63}$ | $41.36_{\pm 0.50}$ | $41.36_{\pm 20.7}$ |
| MEMO (P) | $80.61_{\pm 0.89}$ | $\mathbf{17.82}_{\pm 0.37}$ | $32.92_{\pm 1.07}$ | $65.97_{\pm 0.61}$ | $49.33_{\pm 0.49}$ | $49.33_{\pm 25.1}$ |
| T3A | $76.99_{\pm 0.43}$ | $17.41_{\pm 0.83}$ | $32.07_{\pm 0.81}$ | $64.07_{\pm 2.43}$ | $47.63_{\pm 0.98}$ | $47.63_{\pm 23.9}$ |
| FedTHE (Ours) | $93.00_{\pm 0.19}$ | $13.71_{\pm 0.79}$ | $32.58_{\pm 0.53}$ | $71.23_{\pm 1.54}$ | $51.04_{\pm 2.22}$ | $52.31_{\pm 31.2}$ |
| FedTHE+ (Ours) | $\mathbf{94.62}_{\pm 0.15}$ | $14.44_{\pm 0.66}$ | $\mathbf{33.65}_{\pm 0.18}$ | $\mathbf{74.40}_{\pm 0.84}$ | $\mathbf{54.74}_{\pm 3.04}$ | $\mathbf{54.37}_{\pm 31.8}$ |

Table 6: Numerical results of Figure 5.

| Methods | ResNet20 with GN | | | | | Compact Convolutional Transformer (CCT4) | | | | |
|---|---|---|---|---|---|---|---|---|---|---|
| | Original local test | Corrupted local test | Original OoC local test | Naturally Shifted test | Mixture of test | Original local test | Corrupted local test | Original OoC local test | Naturally Shifted test | Mixture of test |
| FedAvg + FT | $82.31_{\pm 0.59}$ | $66.98_{\pm 0.66}$ | $63.90_{\pm 1.45}$ | $72.60_{\pm 0.99}$ | $71.44_{\pm 0.36}$ | $82.92_{\pm 0.56}$ | $55.50_{\pm 2.02}$ | $62.61_{\pm 1.38}$ | $71.01_{\pm 1.38}$ | $68.01_{\pm 0.74}$ |
| FedRoD | $83.15_{\pm 1.16}$ | $70.75_{\pm 6.23}$ | $62.61_{\pm 1.88}$ | $72.38_{\pm 1.61}$ | $71.31_{\pm 0.50}$ | $82.96_{\pm 0.93}$ | $55.75_{\pm 1.89}$ | $59.72_{\pm 2.14}$ | $70.42_{\pm 1.25}$ | $67.21_{\pm 0.96}$ |
| FedRep | $82.66_{\pm 0.46}$ | $67.29_{\pm 0.89}$ | $61.75_{\pm 1.97}$ | $72.82_{\pm 1.46}$ | $71.13_{\pm 0.03}$ | $83.29_{\pm 0.58}$ | $56.22_{\pm 1.50}$ | $62.14_{\pm 1.80}$ | $71.44_{\pm 1.60}$ | $68.28_{\pm 0.57}$ |
| MEMO (P) | $84.69_{\pm 0.12}$ | $66.76_{\pm 0.78}$ | $65.60_{\pm 1.51}$ | $74.37_{\pm 0.52}$ | $72.84_{\pm 0.36}$ | $83.80_{\pm 0.46}$ | $57.95_{\pm 1.45}$ | $62.53_{\pm 1.83}$ | $72.51_{\pm 1.24}$ | $69.20_{\pm 0.54}$ |
| T3A | $81.45_{\pm 0.53}$ | $65.23_{\pm 0.62}$ | $68.32_{\pm 1.93}$ | $71.76_{\pm 0.58}$ | $71.69_{\pm 0.40}$ | $79.46_{\pm 0.73}$ | $50.17_{\pm 1.85}$ | $64.58_{\pm 1.76}$ | $69.69_{\pm 0.97}$ | $65.99_{\pm 0.36}$ |
| FedTHE (Ours) | $88.37_{\pm 0.92}$ | $72.94_{\pm 0.94}$ | $81.89_{\pm 0.53}$ | $77.08_{\pm 0.95}$ | $78.70_{\pm 0.36}$ | $86.88_{\pm 0.40}$ | $56.35_{\pm 1.30}$ | $\mathbf{76.69}_{\pm 2.06}$ | $73.24_{\pm 0.55}$ | $72.54_{\pm 0.41}$ |
| FedTHE+ (Ours) | $\mathbf{89.27}_{\pm 0.75}$ | $\mathbf{73.86}_{\pm 1.35}$ | $\mathbf{82.66}_{\pm 0.46}$ | $\mathbf{78.75}_{\pm 0.98}$ | $\mathbf{80.06}_{\pm 0.77}$ | $\mathbf{87.97}_{\pm 0.28}$ | $\mathbf{58.28}_{\pm 0.91}$ | $76.54_{\pm 2.17}$ | $\mathbf{74.67}_{\pm 1.00}$ | $\mathbf{73.97}_{\pm 0.56}$ |

Table 7: **The synthetic corrupted out-of-client local test accuracy of representative methods** (i.e. a joint shift of common corruptions and Out-of-Client shift) on different degree of non-i.i.d of local data distribution, for training on CIFAR10.

| Architectures | Dir ($\alpha$) | FedAvg + FT | T3A | FedRep | FedRoD | MEMO (P) | FedTHE (Ours) | FedTHE+ (Ours) |
|---|---|---|---|---|---|---|---|---|
| CCT (w/ LN) | $\alpha = 1.0$ | $38.54 \pm 0.76$ | $37.53 \pm 0.52$ | $37.82 \pm 0.78$ | $37.43 \pm 0.97$ | $38.97 \pm 0.14$ | $45.56 \pm 0.67$ | $\mathbf{46.54} \pm 0.56$ |
| ResNet (w/ GN) | $\alpha = 1.0$ | $46.70 \pm 1.60$ | $49.03 \pm 1.16$ | $46.88 \pm 1.75$ | $46.91 \pm 1.88$ | $65.08 \pm 0.30$ | $65.99 \pm 0.82$ | $45.19 \pm 1.25$ |
| CNN | $\alpha = 0.1$ | $26.84 \pm 1.67$ | $29.54 \pm 1.64$ | $23.07 \pm 1.71$ | $22.98 \pm 1.76$ | $23.02 \pm 1.39$ | $44.68 \pm 2.27$ | $\mathbf{44.78} \pm 1.58$ |
| | $\alpha = 1.0$ | $46.43 \pm 1.39$ | $45.46 \pm 1.26$ | $43.40 \pm 0.92$ | $43.18 \pm 1.53$ | $44.68 \pm 1.22$ | $52.36 \pm 1.16$ | $\mathbf{53.34} \pm 0.73$ |

Table 8: **Comparison of FedTHE and MEMO (P) with different corruption severity** (for training CNN on CIFAR10), where the corruption severity only affects the performance of synthetic corrupted local test and mixture of test.

| Non-i.i.d | Test type | Method | Levels of severity | | | | |
|---|---|---|---|---|---|---|---|
| | | | 1 | 2 | 3 | 4 | 5 |
| Dir(0.1) | Synthetic corrupted local test | MEMO (P) | $87.93_{\pm 0.13}$ | $87.16_{\pm 0.10}$ | $85.79_{\pm 0.21}$ | $84.62_{\pm 0.36}$ | $80.87_{\pm 0.44}$ |
| | | FedTHE (Ours) | $88.62_{\pm 0.39}$ | $87.58_{\pm 0.68}$ | $86.59_{\pm 0.69}$ | $84.58_{\pm 0.53}$ | $81.62_{\pm 0.93}$ |
| | Mixture of test | MEMO (P) | $72.10_{\pm 0.43}$ | $71.84_{\pm 0.46}$ | $71.57_{\pm 0.34}$ | $71.29_{\pm 0.36}$ | $69.73_{\pm 0.46}$ |
| | | FedTHE (Ours) | $73.41_{\pm 0.75}$ | $73.15_{\pm 0.89}$ | $72.94_{\pm 0.68}$ | $72.53_{\pm 0.77}$ | $72.07_{\pm 0.57}$ |
| Dir(1.0) | Synthetic corrupted local test | MEMO (P) | $77.79_{\pm 0.48}$ | $75.82_{\pm 0.84}$ | $73.60_{\pm 0.69}$ | $70.63_{\pm 0.79}$ | $66.86_{\pm 0.82}$ |
| | | FedTHE (Ours) | $78.51_{\pm 1.01}$ | $76.8_{\pm 1.02}$ | $74.57_{\pm 0.91}$ | $71.77_{\pm 1.10}$ | $66.93_{\pm 1.03}$ |
| | Mixture of test | MEMO (P) | $71.16_{\pm 0.26}$ | $70.72_{\pm 0.42}$ | $70.18_{\pm 0.27}$ | $69.43_{\pm 0.32}$ | $68.41_{\pm 0.50}$ |
| | | FedTHE (Ours) | $72.38_{\pm 0.90}$ | $72.02_{\pm 0.99}$ | $71.38_{\pm 1.16}$ | $70.82_{\pm 0.98}$ | $69.88_{\pm 1.19}$ |

**Ablation study for larger group of clients.** In Table 12, we show additional results for a larger group of clients (i.e., 100). Compared to strong baselines, our methods FedTHE and FedTHE+ are still significantly outperform the others on all ID & OOD cases, which desmonstrates that the proposed scheme generalizes well on larger number of clients. We avoid training with 100 cients on ImageNet as it is far beyond the computational feasibility.

**Ablation study for hyperparameters $\alpha$ and $\beta$.** In Figure 16, we take the example of training CNN on CIFAR10 under Dir(0.1) and plot the heat map of performance for different choices of $\alpha$ and $\beta$, for the purpose of illustrating better hyperparameters. We use FedTHE here to decouple the effect of the test-time fine-tuning phase in FedTHE+. For 5 different test distributions (1 ID and 4 OOD data), one observation is that when increasing any of the $\alpha$ and $\beta$, the performance of mixture of test is improved, while for other 4 test distributions, the best choice of $\alpha$ is either 0.05 or 0.01 and is not very dependent on the value of $\beta$. Based on such observation, we finally choose $\alpha = 0.1$ and $\beta = 0.3$

Table 9: **Ablation study of comparing MEMO (P) and FedTHE on different local personalization epochs** (for training CNN on CIFAR10).

| Local personalized training epochs | Methods | Local data distribution: Dir(1.0) | | | | |
| --- | --- | --- | --- | --- | --- | --- |
| | | Original local test | Corrupted local test | Original OoC local test | Naturally Shifted test | Mixture of test |
| 2 | MEMO (P) | $79.69_{\pm 0.46}$ | $66.14_{\pm 0.08}$ | $57.09_{\pm 1.23}$ | $68.61_{\pm 0.50}$ | $67.88_{\pm 0.19}$ |
| | FedTHE | $80.76_{\pm 1.06}$ | $67.28_{\pm 1.11}$ | $67.97_{\pm 0.93}$ | $66.69_{\pm 1.33}$ | $70.07_{\pm 0.90}$ |
| 4 | MEMO (P) | $79.10_{\pm 0.46}$ | $65.26_{\pm 0.08}$ | $55.46_{\pm 1.23}$ | $68.07_{\pm 0.50}$ | $66.97_{\pm 0.19}$ |
| | FedTHE | $80.90_{\pm 1.02}$ | $67.30_{\pm 0.87}$ | $68.42_{\pm 1.29}$ | $66.34_{\pm 1.29}$ | $69.76_{\pm 0.65}$ |
| 8 | MEMO (P) | $78.63_{\pm 0.42}$ | $65.06_{\pm 0.28}$ | $54.10_{\pm 1.61}$ | $67.79_{\pm 1.61}$ | $66.39_{\pm 0.18}$ |
| | FedTHE | $80.74_{\pm 1.01}$ | $67.04_{\pm 0.76}$ | $67.79_{\pm 0.51}$ | $66.42_{\pm 1.37}$ | $69.85_{\pm 0.87}$ |
| | | Local data distribution: Dir(0.1) | | | | |
| 2 | MEMO (P) | $88.67_{\pm 0.42}$ | $81.61_{\pm 0.39}$ | $28.04_{\pm 3.53}$ | $82.56_{\pm 0.45}$ | $70.22_{\pm 0.97}$ |
| | FedTHE | $89.75_{\pm 0.48}$ | $81.78_{\pm 0.66}$ | $57.75_{\pm 2.28}$ | $80.43_{\pm 0.59}$ | $72.27_{\pm 0.32}$ |
| 4 | MEMO (P) | $88.48_{\pm 0.42}$ | $81.26_{\pm 0.39}$ | $27.54_{\pm 3.53}$ | $82.04_{\pm 0.45}$ | $69.83_{\pm 0.97}$ |
| | FedTHE | $89.77_{\pm 0.46}$ | $81.54_{\pm 1.11}$ | $58.37_{\pm 3.09}$ | $80.83_{\pm 0.66}$ | $72.03_{\pm 0.53}$ |
| 8 | MEMO (P) | $88.12_{\pm 0.56}$ | $81.06_{\pm 0.56}$ | $27.41_{\pm 2.82}$ | $81.59_{\pm 1.11}$ | $69.54_{\pm 0.39}$ |
| | FedTHE | $89.83_{\pm 0.41}$ | $81.73_{\pm 1.42}$ | $57.78_{\pm 3.33}$ | $80.25_{\pm 0.80}$ | $71.80_{\pm 0.26}$ |

Table 10: **Ablation study on FedTHE with different sizes of local test set** (for training CNN on CIFAR10). Smaller local test sets, which is of 0.75 and 0.5 times of test samples compared to Table 1, are considered. We didn't evaluate on MEMO (P) in this case because its performance is dependent to the size of local test set (no need to maintain test history as FedTHE and FedTHE+ do).

| Size of local test set | Methods | Local data distribution: Dir(1.0) | | | | |
| --- | --- | --- | --- | --- | --- | --- |
| | | Original local test | Corrupted local test | Original OoC local test | Naturally Shifted test | Mixture of test |
| 1 | | $80.49_{\pm 0.89}$ | $66.93_{\pm 1.03}$ | $68.39_{\pm 0.64}$ | $66.25_{\pm 0.96}$ | $69.88_{\pm 1.19}$ |
| 0.75 | FedTHE | $80.38_{\pm 0.84}$ | $67.57_{\pm 1.13}$ | $67.27_{\pm 0.80}$ | $66.11_{\pm 1.66}$ | $69.47_{\pm 0.62}$ |
| 0.5 | | $80.47_{\pm 0.88}$ | $67.18_{\pm 1.15}$ | $67.74_{\pm 1.08}$ | $65.96_{\pm 1.93}$ | $69.19_{\pm 1.29}$ |
| | | Local data distribution: Dir(0.1) | | | | |
| 1 | | $89.58_{\pm 0.45}$ | $81.62_{\pm 0.93}$ | $57.12_{\pm 3.66}$ | $80.88_{\pm 0.80}$ | $72.07_{\pm 0.57}$ |
| 0.75 | FedTHE | $89.63_{\pm 0.25}$ | $82.09_{\pm 0.89}$ | $57.34_{\pm 2.69}$ | $80.58_{\pm 0.8}$ | $72.18_{\pm 0.68}$ |
| 0.5 | | $89.84_{\pm 0.16}$ | $81.84_{\pm 1.04}$ | $57.08_{\pm 3.17}$ | $80.56_{\pm 0.75}$ | $71.96_{\pm 0.42}$ |

Table 11: **Ablation study on FedTHE with different sizes of local training set** (for training CNN on CIFAR10). Smaller local training sets, which is of 0.75 and 0.5 times of training samples compared to Table 1, are considered.

| Size of local training set | Methods | Local data distribution: Dir(1.0) | | | | |
| --- | --- | --- | --- | --- | --- | --- |
| | | Original local test | Corrupted local test | Original OoC local test | Naturally Shifted test | Mixture of test |
| 1 | MEMO (P) | $80.20_{\pm 0.69}$ | $66.86_{\pm 0.82}$ | $57.80_{\pm 2.44}$ | $68.79_{\pm 1.46}$ | $68.41_{\pm 0.50}$ |
| | FedTHE | $80.49_{\pm 0.89}$ | $66.93_{\pm 1.03}$ | $68.39_{\pm 0.64}$ | $66.25_{\pm 0.96}$ | $69.88_{\pm 1.19}$ |
| 0.75 | MEMO (P) | $74.35_{\pm 1.36}$ | $63.66_{\pm 1.02}$ | $59.42_{\pm 1.40}$ | $61.40_{\pm 2.41}$ | $63.87_{\pm 0.75}$ |
| | FedTHE | $77.94_{\pm 1.27}$ | $65.79_{\pm 1.19}$ | $64.58_{\pm 1.43}$ | $64.28_{\pm 1.70}$ | $67.58_{\pm 0.65}$ |
| 0.5 | MEMO (P) | $72.62_{\pm 0.86}$ | $61.96_{\pm 0.40}$ | $48.88_{\pm 0.44}$ | $62.35_{\pm 1.93}$ | $61.45_{\pm 0.72}$ |
| | FedTHE | $76.49_{\pm 1.06}$ | $63.92_{\pm 0.27}$ | $52.56_{\pm 1.44}$ | $66.28_{\pm 0.73}$ | $64.81_{\pm 0.15}$ |
| | | Local data distribution: Dir(0.1) | | | | |
| 1 | MEMO (P) | $88.19_{\pm 0.79}$ | $80.87_{\pm 0.44}$ | $28.25_{\pm 1.96}$ | $81.61_{\pm 0.51}$ | $69.73_{\pm 0.46}$ |
| | FedTHE | $89.58_{\pm 0.45}$ | $81.62_{\pm 0.93}$ | $57.12_{\pm 3.66}$ | $80.88_{\pm 0.80}$ | $72.07_{\pm 0.57}$ |
| 0.75 | MEMO (P) | $86.06_{\pm 0.63}$ | $79.13_{\pm 0.78}$ | $49.43_{\pm 4.18}$ | $77.61_{\pm 0.77}$ | $68.53_{\pm 0.74}$ |
| | FedTHE | $87.95_{\pm 0.34}$ | $80.48_{\pm 0.97}$ | $54.13_{\pm 2.69}$ | $79.02_{\pm 1.07}$ | $70.20_{\pm 0.26}$ |
| 0.5 | MEMO (P) | $83.06_{\pm 0.72}$ | $77.38_{\pm 1.06}$ | $25.09_{\pm 2.94}$ | $77.76_{\pm 1.74}$ | $65.57_{\pm 0.15}$ |
| | FedTHE | $86.58_{\pm 0.55}$ | $78.69_{\pm 1.42}$ | $26.01_{\pm 3.18}$ | $79.30_{\pm 1.84}$ | $67.64_{\pm 1.60}$ |

as the best trade-off point for the overall performance, and use these hyperparameters throughout all the other experiments.

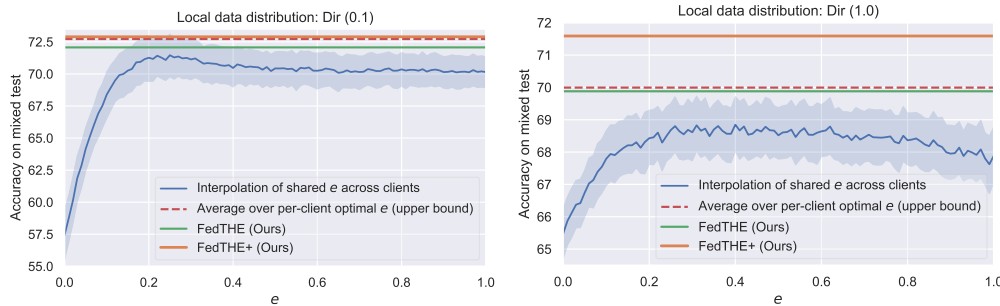

Figure 14: **Ablation study on different strategies to interpolate local personalized and global generic head** (training CNN on CIFAR10), as an additional comparison with other prior personalized FL methods (Deng et al., 2020a; Mansour et al., 2020). Two strategies have been evaluated, but both fall short of ours: **(left)** a shared $e$ across clients is used for interpolation, and **(right)** an unrealistic theoretical upper bound where each client uses its optimal $e$ for interpolation. ($e$ denotes ensemble weight for global model)

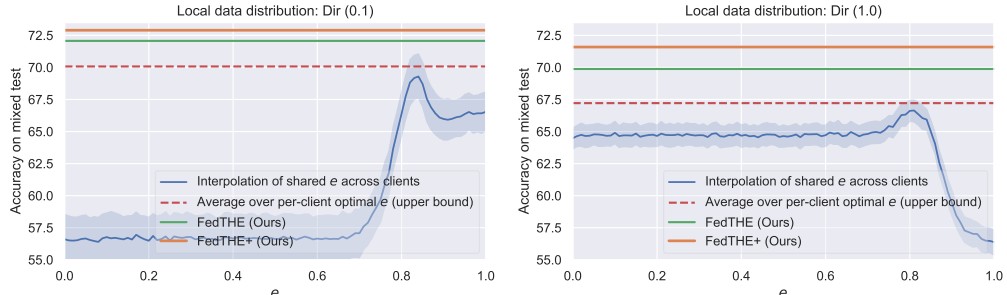

Figure 15: **Ablation study on different strategies to interpolate local personalized and global generic model** (training CNN on CIFAR10), as an additional comparison with other prior personalized FL methods (Deng et al., 2020a; Mansour et al., 2020). Two strategies have been evaluated, but both fall short of ours: **(left)** a shared $e$ across clients is used for interpolation, and **(right)** an unrealistic theoretical upper bound where each client uses its optimal $e$ for interpolation. Similar findings can be observed in Figure 14 (of Appendix E.4) for head-only interpolation. ($e$ denotes ensemble weight for global model)

Table 12: **Training on CIFAR10 with 100 clients.** Selected strong baselines (based on Table 1) and our methods (FedTHE and FedTHE+) are compared on Dir(0.1).

| Methods | Local data distribution: Dir(0.1) | | | | |
|---|---|---|---|---|---|
| | Original local test | Corrupted local test | Original OoC local test | Naturally Shifted test | Mixture of test |
| FedAvg + FT | $83.07_{\pm 1.11}$ | $76.24_{\pm 1.12}$ | $41.75_{\pm 0.67}$ | $76.83_{\pm 0.43}$ | $69.48_{\pm 0.48}$ |
| FedRoD | $87.07_{\pm 0.94}$ | $79.00_{\pm 0.23}$ | $29.61_{\pm 1.52}$ | $78.90_{\pm 0.27}$ | $68.60_{\pm 0.09}$ |
| MEMO (P) | $83.79._{\pm 1.77}$ | $76.39_{\pm 1.82}$ | $37.87_{\pm 0.52}$ | $78.62_{\pm 0.58}$ | $69.17_{\pm 0.92}$ |
| FedTHE (Ours) | $88.55_{\pm 0.29}$ | $78.15_{\pm 1.25}$ | $55.96_{\pm 0.73}$ | $77.86_{\pm 0.03}$ | $74.46_{\pm 0.27}$ |
| FedTHE+ (Ours) | $\mathbf{89.00}_{\pm 0.05}$ | $\mathbf{79.38}_{\pm 0.42}$ | $\mathbf{57.48}_{\pm 0.87}$ | $\mathbf{79.10}_{\pm 0.22}$ | $\mathbf{75.43}_{\pm 0.05}$ |

Although we choose 0.1 and 0.3 as our better trade-off point, it is worth noticing that within the tested range of $\alpha$ and $\beta$, FedTHE always gives relatively good results, indicating that the method is not very sensitive to the choice of hyperparameters, which is a favourable property for online deployment.

**Ablation study for client sampling ratio.** In our experiments, we consider 20 & 100 clients for CIFAR10 and 20 clients for ImageNet, and we discussed the reason for our choice in subsection C.1. Upon extensively taking various distribution shifts & data heterogeneity and tuning hyperparameters for each baseline, the client sampling ratio is set to 1 to disentangle the effect of sampling and simplify experimental setup. As shown in Table 14 and Table 15, we further provide experimental results for strong baselines in Table 1, with low / medium sampling ratio (0.1 / 0.4). Our method can still consistently outperform others.

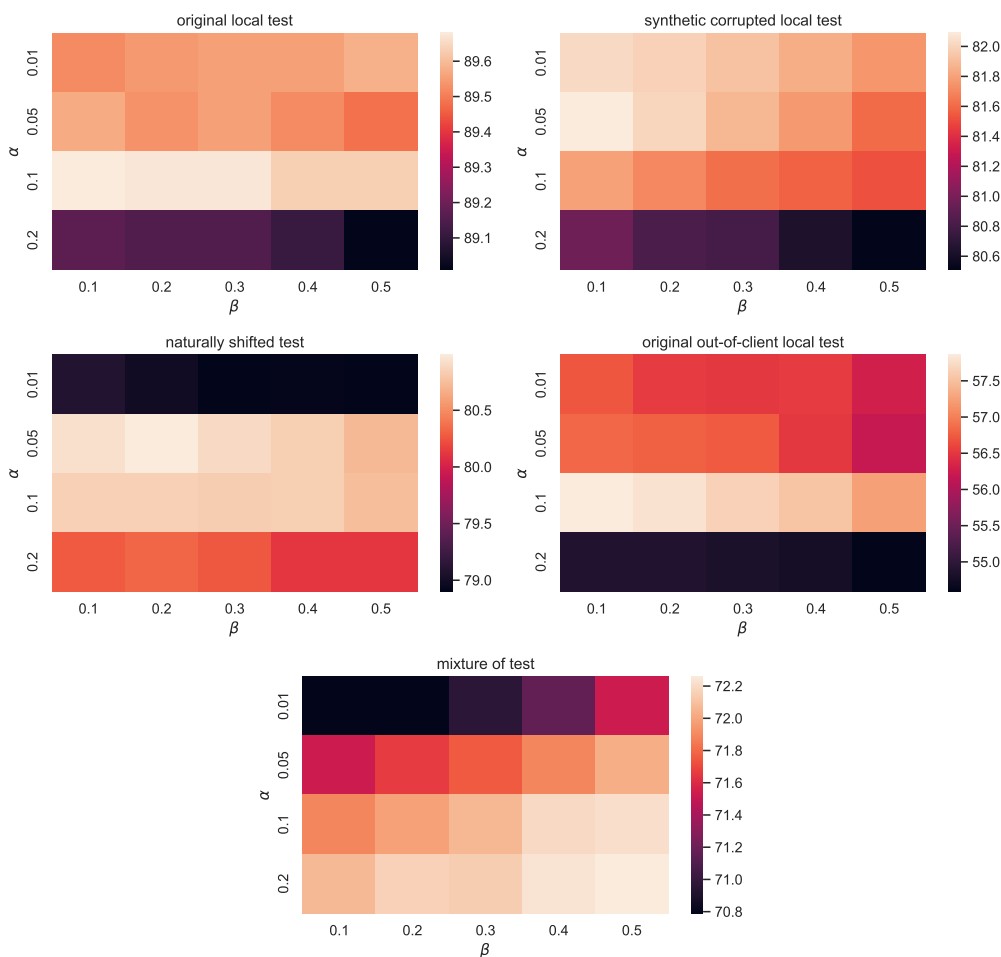

Figure 16: Effect of different $\alpha$ and $\beta$ on FedTHE (training CNN on CIFAR10) with different test distributions and Dir(0.1).

Table 13: **Ablation study for Balanced Softmax** (training CNN on CIFAR10 with Dir(0.1)). The indentation with different symbols denotes adding (+) / removing (–) a component, or using a variant (○).

| Design choices | Methods | Original local test | Corrupted local test | Original OoC local test | Naturally Shifted test | Mixture of test | Average |
|---|---|---|---|---|---|---|---|
| ① | MEMO (P): a strong baseline | 88.19 ±0.79 | 80.87 ±0.44 | 28.25 ±1.96 | 81.61 ±0.51 | 69.73 ±0.46 | 69.73 ±24.1 |
| ② | + test-time LP-FT | 88.52 ±0.41 | 81.51 ±0.83 | 28.62 ±2.63 | 81.75 ±0.81 | 70.10 ±0.35 | 70.10 ±21.6 |
| ③ | TSA on our pure 2-head model | 87.68 ±0.62 | 79.38 ±0.87 | 32.38 ±1.70 | 79.21 ±1.95 | 69.66 ±0.44 | 69.66 ±21.8 |
| ⑮ | FedAvg + FT + BalancedSoftmax | 87.40 ±0.66 | 80.26 ±1.84 | 33.08 ±3.31 | 80.28 ±2.16 | 70.25 ±0.65 | 70.25 ±25.01 |
| ⑯ | + BalancedSoftmax on personalized haead | 68.53 ±2.69 | 56.05 ±2.37 | 62.67 ±1.48 | 54.59 ±2.40 | 60.46 ±2.05 | 60.46 ±5.6 |
| ④ | Our pure 2-head model | 87.49 ±0.17 | 79.74 ±0.70 | 33.34 ±2.46 | 79.17 ±1.36 | 69.90 ±0.15 | 69.90 ±21.4 |
| ⑤ | + BalancedSoftmax | 88.68 ±0.17 | 81.11 ±0.59 | 32.92 ±2.55 | 80.18 ±1.00 | 70.72 ±0.36 | 70.72 ±22.1 |
| ⑥ | + FA | 87.41 ±0.35 | 76.10 ±1.12 | **63.39** ±2.87 | 77.00 ±1.47 | 67.98 ±0.72 | 74.38 ±9.2 |
| ⑦ | + EM | 71.37 ±1.15 | 60.54 ±0.66 | 49.62 ±0.45 | 57.38 ±1.40 | 59.98 ±1.23 | 59.78 ±7.8 |
| ⑧ | + Both (i.e. 0.5 FA + 0.5 EM) | 88.54 ±0.43 | 81.40 ±0.47 | 41.45 ±2.51 | 80.55 ±0.57 | 70.57 ±0.79 | 72.50 ±18.5 |
| ⑨ | + SLW (i.e. FedTHE) | 89.58 ±0.45 | 81.62 ±0.93 | 57.12 ±3.66 | 80.88 ±0.80 | 72.07 ±0.57 | 76.25 ±12.4 |
| ⑩ | ○ Batch-wise FedTHE | 89.97 ±0.24 | 82.58 ±1.21 | 54.12 ±1.79 | 81.58 ±1.06 | 70.10 ±0.26 | 75.67 ±14.0 |
| ⑪ | – Test history $\mathbf{h}^{history}$ | 87.05 ±0.68 | 77.79 ±1.23 | 50.45 ±2.10 | 76.99 ±0.98 | 71.89 ±1.28 | 72.83 ±13.7 |
| ⑫ | – BalancedSoftmax | 89.29 ±0.75 | 81.32 ±0.18 | 56.46 ±1.84 | 80.65 ±1.08 | 71.81 ±0.95 | 75.91 ±12.5 |
| ⑬ | + FT (i.e. FedTHE+) | **90.55** ±0.41 | **82.71** ±0.69 | 58.29 ±2.05 | **81.91** ±0.54 | **72.90** ±0.93 | **77.27** ±12.3 |
| ⑭ | – BalancedSoftmax | 90.32 ±0.69 | 82.86 ±0.90 | 58.19 ±3.81 | 81.74 ±0.53 | 72.33 ±1.22 | 77.09 ±12.4 |

Table 14: **Training on CIFAR10 with 20 clients with sampling ratio 0.1 and 100 communication rounds.** Selected strong baselines (based on Table 1) and our methods (FedTHE and FedTHE+) are compared on Dir(0.1).

| Methods | Sampling Ratio (0.1)) | | | | |
|---|---|---|---|---|---|
| | Original local test | Corrupted local test | Original OoC local test | Naturally Shifted test | Mixture of test |
| FedAvg + FT | 87.13 ±0.97 | 79.72 ±1.07 | 30.92 ±3.12 | 79.50 ±2.10 | 69.32 ±0.73 |
| FedRoD | 88.23 ±0.66 | 81.35 ±0.82 | 26.80 ±2.47 | 81.47 ±1.63 | 69.46 ±0.15 |
| kNN-Per | 87.29 ±1.16 | 78.30 ±0.66 | 25.90 ±4.51 | 80.62 ±1.35 | 68.03 ±1.10 |
| MEMO (P) | 88.34 ±0.54 | 81.01 ±0.27 | 27.52 ±3.27 | 81.42 ±1.66 | 69.57 ±0.50 |
| FedTHE (Ours) | 88.70 ±0.71 | 81.15 ±0.44 | 53.58 ±2.69 | 80.85 ±0.69 | 70.07 ±0.93 |
| FedTHE+ (Ours) | **89.48** ±0.58 | **81.98** ±0.66 | **53.69** ±2.70 | **81.66** ±0.47 | **70.40** ±0.69 |

Table 15: **Training on CIFAR10 with 20 clients with sampling ratio 0.4 and 100 cmmunication rounds.** Selected strong baselines (based on Table 1) and our methods (FedTHE and FedTHE+) are compared on Dir(0.1).

| Methods | Sampling Ratio (0.4)) | | | | |
|---|---|---|---|---|---|
| | Original local test | Corrupted local test | Original OoC local test | Naturally Shifted test | Mixture of test |
| FedAvg + FT | 87.58 ±0.77 | 80.53 ±0.96 | 32.34 ±2.39 | 80.80 ±1.62 | 70.31 ±0.32 |
| FedRoD | 88.77 ±0.53 | 81.76 ±1.11 | 27.14 ±2.50 | 81.30 ±1.27 | 69.74 ±0.11 |
| kNN-Per | 88.92 ±0.67 | 80.40 ±0.37 | 30.49 ±4.52 | 81.23 ±1.43 | 70.26 ±0.72 |
| MEMO (P) | 88.77 ±0.20 | 81.86 ±0.35 | 28.59 ±2.84 | 81.49 ±1.13 | 70.18 ±0.35 |
| FedTHE (Ours) | 89.21 ±0.55 | 81.87 ±1.11 | 57.80 ±3.52 | 80.71 ±1.33 | 71.54 ±0.69 |
| FedTHE+ (Ours) | **89.91** ±0.46 | **82.16** ±1.16 | **58.03** ±3.42 | **81.64** ±0.72 | **71.79** ±0.74 |

## E.5 MORE ANALYSIS ON HEAD ENSEMBLE WEIGHT $e$

Here we provide visualization and analysis on head ensemble weight $e$, which is the key of our proposed FedTHE, where in Figure 17 we visualize the distribution (probability density) of $1 - e$ (which is the ensemble weight for the local head) across various test distributions including 1 ID and 4 OOD distributions. Note that each $1 - e$ corresponds to a single test sample from the corresponding test distribution. We can see that:

- For in-distribution case: for most samples (major classes), large weights on the local head are learned since the local head fits local distribution; for a very small portion of samples that belong to tailed local classes, a smaller ensemble weight on local head is learned, in order to better utilize the knowledge from the global head.

- For in-distribution / common corruptions / natural distribution shift cases, the distribution of $1-e$ reflects the insights in "Accuracy-on-the-line" (Miller et al., 2021), thus similar patterns

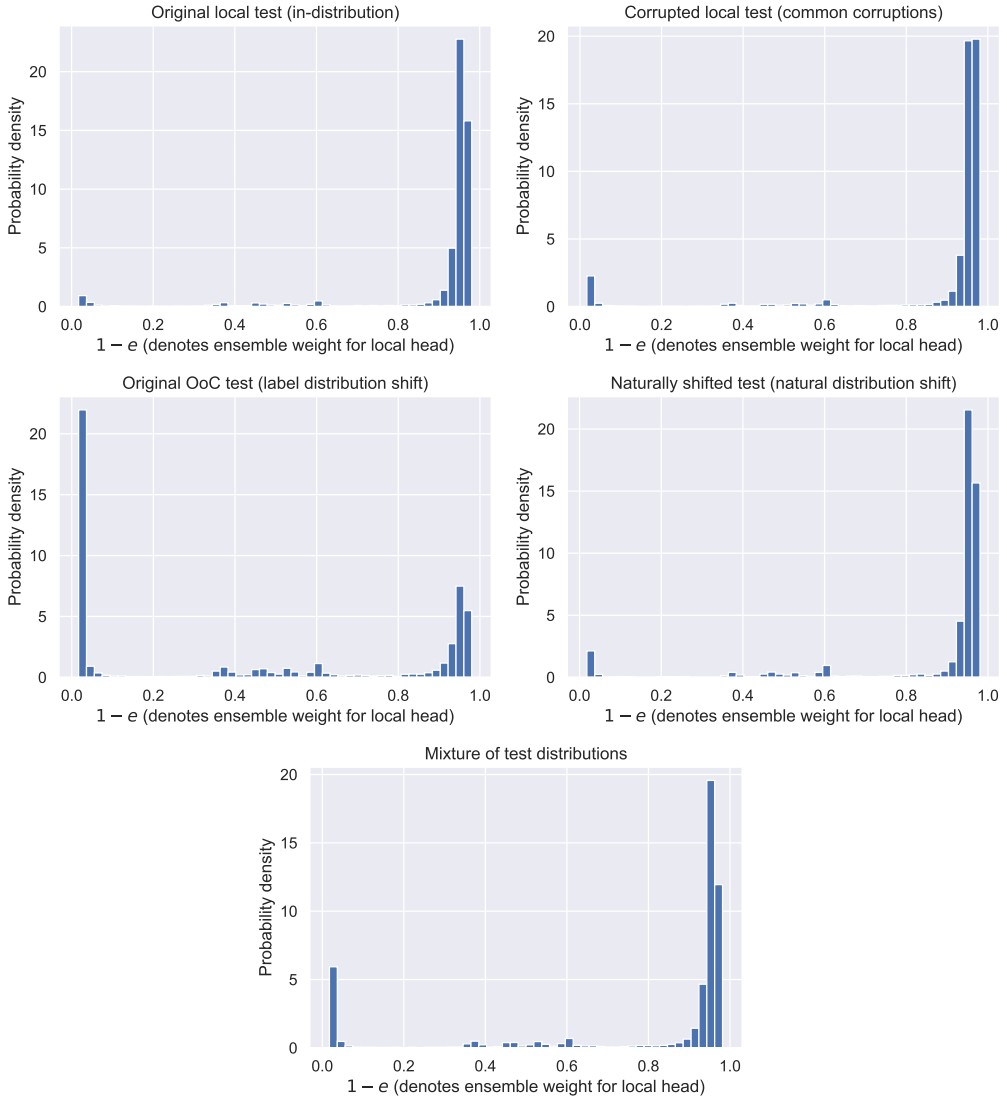

Figure 17: The distribution (probability density) of $1 - e$ across clients.

with some differences (e.g. the usage of global model increases in common corruptions and natural distribution shift) can be observed.

- For label distribution shift case, our method learns to utilize global head as the main power of prediction when encountering tailed or unseen classes, while for a smaller portion of test samples whose labels fall in the present set of local training labels, the local head is favored.

### E.6 MORE ANALYSIS ON THE EFFECTS OF $\mathbf{h}^{\text{history}}$

As a variance reduction is quite important in per-sample adaptation, $\mathbf{h}^{\text{history}}$ and the moving average are supposed to stabilize the choice of $e$ in FedTHE and FedTHE+. We demonstrate the effectiveness and effects of $\mathbf{h}^{\text{history}}$ here. As shown in Figure 18, we visualize the feature representation with and without $\mathbf{h}^{\text{history}}$ with t-SNE in 2-dimension, and when history is considered, the clusters are more concentrated, and the effects of this concentrated pattern are discussed below. For the effects of history, we show that, in Figure 19, the history makes a difference on choosing a proper $e$. Accompanying with the ⑪ in Table 3, we see that such difference is crucial on the preformance of the method.

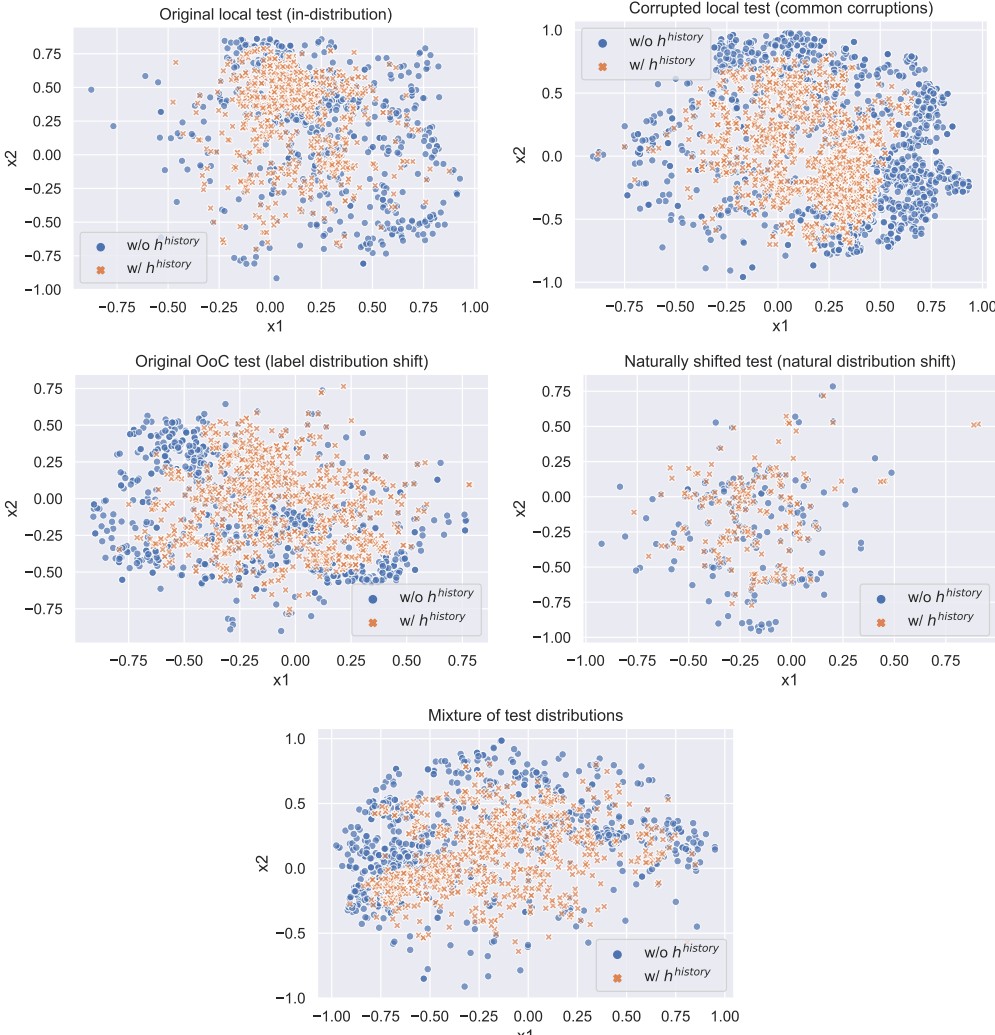

Figure 18: The distribution (probability density) of $1 - e$ across clients.

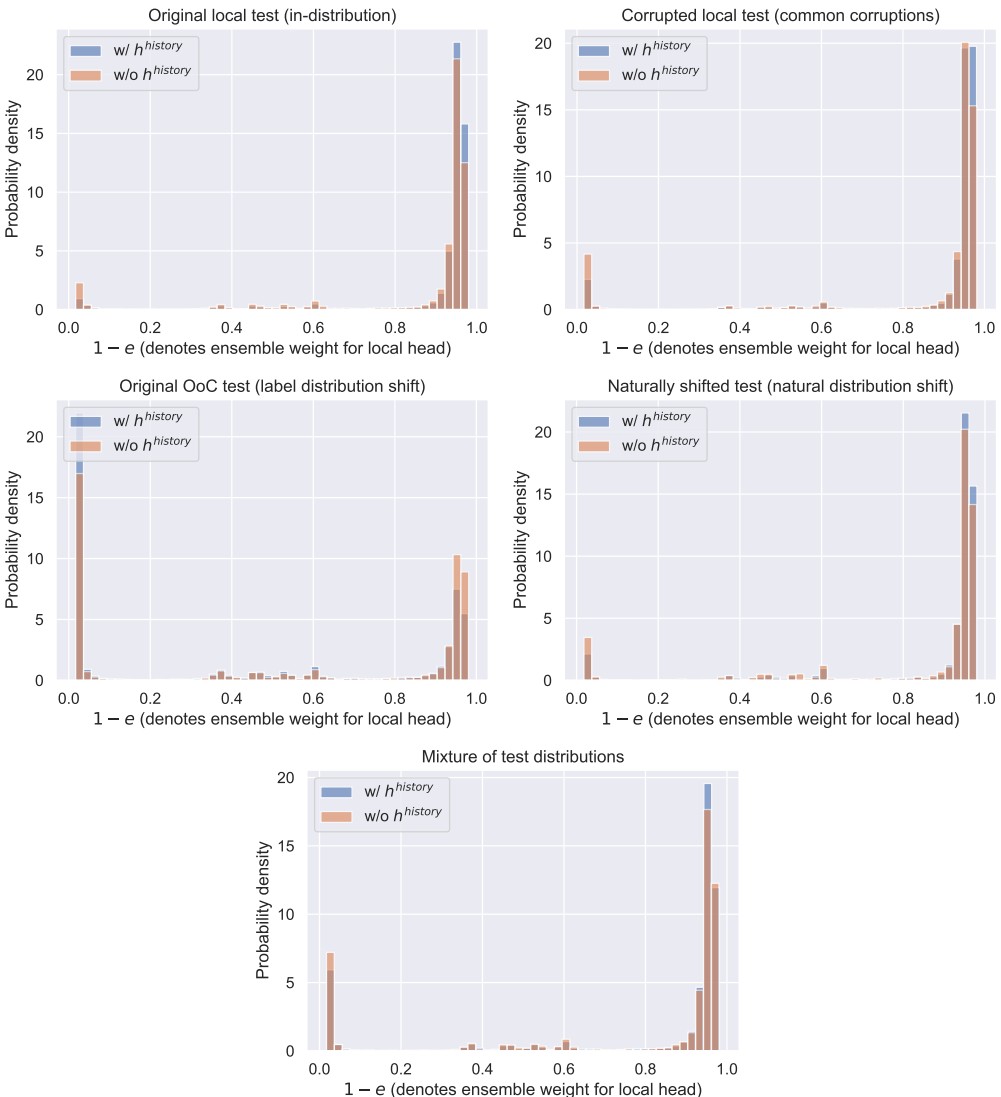

Figure 19: The distribution (probability density) of $1 - e$ across clients.

