# OpenReview forum: "Test-Time Robust Personalization for Federated Learning"
_ICLR.cc/2023/Conference — ICLR 2023 poster_

### Official Review · Reviewer_129X · 2022-10-24

**Confidence:** 4
**Correctness:** 3
**Technical Novelty And Significance:** 3
**Empirical Novelty And Significance:** 3
**Recommendation:** 8

**Clarity, Quality, Novelty And Reproducibility:**

This paper is mostly clear and well written. One minor comment is that there seems to be a bit excessive use of \vspace that makes the text look cramped, so I would suggest for some things to be moved to the appendix so that the paper looks better. Apart from that, another minor comment is that MEMO (G) needs to be bold at Table 1 for Dir(0.1), OoC local test and Dir(1.0) OoC local test.

While this work does have a lot of moving parts, the authors do extensive experimental evaluation and ablation studies to show the effects of each one. Besides that, the authors also perform an extensive hyperparameter optimization procedure for the baselines as well. I believe that there is enough information in this work for reproducibility.

As far as I am aware, FedTHE is novel. Furthermore, its simplicity coupled with the extensive distribution shifts considered, makes it quite appealing for FL type of scenarios. One nitpick on the loss is that the “feature alignment” is a bit misleading, as the features are frozen. Something like adapting the scalar according to the feature distance would be more appropriate. The similarity-guided loss weighting is also an interesting and intuitive contribution. FedTHE+ is not novel per se, as the authors just use existing test-time adaptation methods (i.e., MEMO) to fine-tune the model. One catch for FedTHE though is that it needs to send summary statistics of the client data to the server ($h_l$), which I believe should be discussed in the context of the baselines (e.g., do baselines employ such summary statistics?) and in the context of the privacy / security aspects of FL.



**Strength And Weaknesses:**

Strengths
- Extensive evaluation and ablation studies
- Hyperparameter tuning for the baselines as well
- Simple method and computationally efficient to apply, which is an important aspect for FL
- Good results

Weaknesses
- A lot of moving parts
- The motivation behind some choices is a bit unclear

**Summary Of The Paper:**

This work introduces FedTHE(+), a way to allow for test-time personalization of federated learning models. FedTHE works by introducing and training a scalar that interpolates global and local classifier predictions. This scalar is tuned at test time by minimising the entropy of the interpolated logits while being regularised by an alignment loss between the features. The latter giudes the scalar according to the difference of the representation of the test point to the local and global feature means; if the representation is closer to the global one it increases the scalar otherwise it decreases it. To balance the two losses, the authors propose to use the cosine similarity between the local and global logits as a “weight” that linearly interpolates the two. After this specific scalar is optimized, the other parameters of the network are also fine-tuned using an entropy-minimisation method, which constitutes the FedTHE+.


**Summary Of The Review:**

This is an interesting work and discusses a relatively simple method FedTHE and its extension FedTHE+. FedTHE is novel and the experimental evaluation is quite extensive, including multiple distribution shift scenarios, with good results and the authors do extensive ablations to show the effects of each of their choices in FedTHE, which is appreciated. For this reason, I am recommending acceptance of this work.

---

> ### Author Response · Authors · 2022-11-11
> **Response to Reviewer 129X**
>
> Dear Reviewer 129X,
>
> Thank you for the acknowledgment of our work and insightful suggestions for improving the paper!
>
> > Response to the comment: This paper is mostly clear and well written. One minor comment is that there seems to be a bit excessive use of \vspace that makes the text look cramped, so I would suggest for some things to be moved to the appendix so that the paper looks better. Apart from that, another minor comment is that MEMO (G) needs to be bold at Table 1 for Dir(0.1), OoC local test and Dir(1.0) OoC local test.
>
> Thank you for pointing out the typo, and we have corrected the bold numbers in Table 1.  We will move less important parts to the appendix to better fit the page limit in the next revision.
>
> > Response to the comment: One nitpick on the loss is that the “feature alignment” is a bit misleading, as the features are frozen. Something like adapting the scalar according to the feature distance would be more appropriate.
>
> Thank you very much for the suggestion. We are considering a better name for it.
>
> > Response to the comment: One catch for FedTHE though is that it needs to send summary statistics of the client data to the server (hl), which I believe should be discussed in the context of the baselines (e.g., do baselines employ such summary statistics?) and in the context of the privacy / security aspects of FL.
>
> To the best of our knowledge, existing baselines do not employ such summary statistics, and we will add a discussion to the paper about this point. As for the privacy/security perspective, we will add a discussion on limitations and negative impacts for the next revision.

---

### Official Review · Reviewer_bhMe · 2022-10-25

**Confidence:** 5
**Clarity, Quality, Novelty And Reproducibility:** See above.
**Correctness:** 3
**Technical Novelty And Significance:** 3
**Empirical Novelty And Significance:** 3
**Recommendation:** 6

**Strength And Weaknesses:**

# Pros
1. The idea of using test-time-training to improve the model robustnees in FL is novel and effective.
2. The empirical results are comprehensive and can validate the superiority of the proposed method.
# Cons
1. Acctually, this paper is not directly for OOD generalization, i.e., the local and global training is still based on standard FedAVG. There are some previous work [1,2] focus on optimizing the worst-case local loss needed to be discussed. In addition, is it possible to train different prediction heads by distributionnaly robust optimization?
2. There are many critical hyperparameters need to be tuned.
# Comments
1. It seems that this paper is a re-submission of NIPS, can the author provide the critical difference between the current and previous version?
# Refs
[1] Distributionally Robust Federated Averaging.
[2] DRFLM: Distributionally Robust Federated Learning with Inter-client Noise via Local Mixup

**Summary Of The Paper:**

This paper studies the OOD generalization problem in the FL setting. Specifically, the author presents to use a test-time-training based strategy to ensemble the global and local prediction heads. The object function use in the TTT phase consists of two parts (1) the entropy of the predictive distribution. (2) feature space constrains. The empirical results have shown significant improvements over various baseline models.

**Summary Of The Review:**

See above.

---

> ### Author Response · Authors · 2022-11-11
> **Response to Reviewer bhMe (1/2)**
>
> Dear Reviewer bhMe,
>
> Thank you very much for the review and feedback, we kindly address your questions as follows.
>
> > Response to the comment: Actually, this paper is not directly for OOD generalization, i.e., the local and global training is still based on standard FedAVG. There are some previous work [1,2] focus on optimizing the worst-case local loss needed to be discussed. In addition, is it possible to train different prediction heads by distributionnaly robust optimization?
>
> We kindly clarify that:
>
> - As discussed in Section 2 `OOD generalization in FL` part, the existing FL OOD generalization works merely contribute a distributionally robust **global** model in order to tackle inter-client data heterogeneity, and we already discussed AFL [3], which first introduces the min-max distributionally robust optimization (DRO) to FL. The [1,2] follow AFL on communication efficiency and noisy training data perspective, respectively.
>     - Such strategies ([1,2,3]) only improve the global model in order to handle inter-client heterogeneity and naturally cannot achieve decent local accuracy when compared to personalized FL methods, and cannot handle intra-client distribution shifts.
>     - As we mentioned in the `discussion` part of Section 4.2, such strategies for improving the global model are complementary, compatible, and orthogonal to our method.
> - With the above point, **our standpoint is orthogonal to the above line of research but still lies in OOD generalization for FL**: (i) We aim to do robust **personalization** and tackle test-time distribution shift; (ii)To the best of our knowledge, we are the first to further identify the intra-client distribution shift problem upon inter-client data heterogeneity. Such scenarios are realistic and meaningful as acknowledged by all reviewers, and our method is effective and efficient on diverse distribution shifts, as well as offering better clean test accuracy (in-distribution local test accuracy) than other personalized FL methods.
> - **Numerical results 1**: Furthermore, in our experiments, we compared with GMA [4] and Ditto [5], where GMA aims to learn invariant mechanisms across clients (i.e. learn a better global model, similar to AFL and [1, 2]), and Ditto, a personalized FL method that enforces uniformity of performance across clients. They share similar intuition as AFL and [1,2] but our method shows significant improvement compared to these baselines.
> - **Numerical results 2**: To further address your concerns, we also add results for DRFA [1] below (on CIFAR10, Dir(0.1), sampling ratio = 1, 20 clients, the learning rate of $\lambda$ is tuned to be 0.01), using official source code. Since DRO methods still rely on a **single** fixed model for clients, they cannot achieve impressive accuracy on each client and also fall short of the intra-client distribution shift problem we investigate in our manuscript.
>
> | Methods | Original local test | Corrupted local test | Original OoC local test | Naturally Shifted test | Mixture of test |
> | --- | --- | --- | --- | --- | --- |
> | DRFA | 64.39$_{\pm 1.37}$ | 52.06$_{\pm 1.42}$ | 63.98$_{\pm 1.87}$ | 50.26$_{\pm 1.46}$ | 57.67$_{\pm 1.52}$ |
> | DRFA + FT | 87.53$_{\pm 0.14}$ | 80.54$_{\pm 0.77}$ | 32.38$_{\pm 2.60}$ | 80.62$_{\pm 1.56}$ | 70.27$_{\pm  0.14}$ |
> | FedTHE (Ours) | 89.58$_{\pm 0.45}$ | 81.62$_{\pm 0.93}$ | 57.12$_{\pm 3.66}$ | 80.88$_{\pm 0.8}$ | 72.07$_{\pm 0.57}$ |
> | FedTHE+ (Ours) | **90.55**$_{\pm 0.41}$ | **82.71**$_{\pm 0.69}$ | **58.29**$_{\pm 2.05}$ | **81.91**$_{\pm 0.54}$ | **72.90**$_{\pm 0.93}$ |
>
> Finally, we have added [1,2] to related work in our revised version. Also, we think that training different prediction heads or deriving new FL personalization schemes via DRO is a promising direction and we wish to explore it in future work.
>
> [3] Mohri, Mehryar, Gary Sivek, and Ananda Theertha Suresh. "Agnostic federated learning." *International Conference on Machine Learning*. PMLR, 2019.
>
> [4] Tenison, Irene, et al. "Gradient Masked Averaging for Federated Learning." *arXiv preprint arXiv:2201.11986* (2022).
>
> [5] Li, Tian, et al. "Ditto: Fair and robust federated learning through personalization." *International Conference on Machine Learning*. PMLR, 2021.

---

> > ### Author Response · Authors · 2022-11-11
> > **Response to Reviewer bhMe (2/2)**
> >
> > > Response to the comment: There are many critical hyperparameters need to be tuned.
> >
> > It is worth emphasizing the advantages of being tuning-free at Inference time. And we would like to highlight that, as claimed in the `Enforcing feature space alignment` and `Discussion` parts of Section 4.2, **our method requires marginal effort to tune and is not sensitive to hyperparameters**:
> >
> > - The only hyperparameters in FedTHE are $\alpha$ and $\beta$ in the feature alignment, which do **not** require tuning. We used **fixed** $\alpha=0.1$ and $\beta=0.3$ throughout all experiments (indicating that they are generalizable across models/datasets/data heterogeneity) and showed in Appendix E.4 (Figure 16) that such hyperparameters are stable in terms of accuracy.
> > - Also note that while using the **fixed** hyperparameters for our method, we tuned hyperparameters for other methods (see Appendix C.2), and our method can still outperform significantly other baselines (see Table 1).
> > - And we also propose similarity-guided loss weighting to **completely save tuning efforts** of the two loss terms (EM & FA), which is one of the interesting contributions as acknowledged by Reviewer 129X.
> >
> > > Response to the question: It seems that this paper is a re-submission of NIPS, can the author provide the critical difference between the current and previous versions?
> >
> > Yes. In the previous version, all reviewers acknowledged the FL OoD robustness is a realistic and interesting direction, and our solution is efficient, effective, and novel. Some concerns were that the previous version was hard to follow. The major updates in this version are that we extensively reorganized the manuscript and added more clarification (figures & texts such as Figure 4), discussions (such as in Section 4.2), and remarks (such as pages 9 and 21), made a better investigation of recent related works and, on the experimental side, we added experiments for the large number of clients and added recent SOTA method (kNN-Per) as a baseline to verify the effectiveness of our method.
> >
> > We hope the above answers address your concerns, and please don't hesitate to let us know if you have any remaining questions.

---

> ### Author Response · Authors · 2022-11-16
> **Hoping that our response could address your concern**
>
> Dear Reviewer bhMe,
>
> Thank you again for your time and effort in reviewing our work! We would appreciate it if you can let us know if our response has addressed your concern and thus improved your assessment of our paper. We look forward to hearing from you and remain at your disposal for any further clarification that you might require.

---

### Official Review · Reviewer_wWXU · 2022-10-26

**Confidence:** 3
**Correctness:** 3
**Technical Novelty And Significance:** 3
**Empirical Novelty And Significance:** 3
**Recommendation:** 5

**Clarity, Quality, Novelty And Reproducibility:**

Due to the page limit, the algorithm is not precisely described in the main text. Many detailed steps are deferred to the appendix.

They provide the source code, although I have not tried to run the code.

**Strength And Weaknesses:**

Strengths

-This paper combines test time training and federated learning, which is an interesting direction.

-The authors provide a benchmark for FL robustness

-The proposed algorithm, FedTHE+, outperforms baselines in most of the scenarios.


Weakness

- The authors do not clearly explain why each step of the algorithm is required and for what reason the algorithm can be good for FL.

- FedTHE, the proposed algorithm, consists of many parts. Many parts have synergy between them, and the synergy produces good performances. However, such a complicated structure makes it difficult to understand which part is the key to dealing with distribution shifts.

- Since this paper has too much content with a page limit, many details are omitted in the main body and deferred to the appendix. So it took a lot of work to read.

**Summary Of The Paper:**

This paper considers the federated learning problem under the test-time distribution shift. Federated learning is a distributed learning paradigm with a server and many clients such that each client has its private data, and the server aggregates the trained model from clients without revealing any private information. Many previous works focused on the performance of the server model, but recently personalization on FL has been studied. However, most works assume distribution matching between training and testing, whereas, in practice, the class distribution can change, unseen classes can appear, and common corruptions or natural distribution shifts can happen. Therefore, it is crucial to design robust FL algorithms for dealing with such distribution shifts. In this work, the authors propose Federated Test-time Head Ensemble plus tuning (FedTHE+), which personalizes FL models robustly to the distribution shifts. They evaluate FedTHE+ with various neural architectures on CIFAR10 and ImageNet with diverse test distributions.

**Summary Of The Review:**

This paper suggests an interesting direction, FL robustness. However, this paper does not provide a good intuition for the topic.

---

> ### Author Response · Authors · 2022-11-11
> **Response to Reviewer wWXU (1/2)**
>
> Dear Reviewer wWXU,
>
> We would like to thank you for the feedback, and we are happy to hear that the proposed direction is interesting. Below we provide responses to your comments.
>
> > Response to the comment: The authors do not clearly explain why each step of the algorithm is required and for what reason the algorithm can be good for FL.
>
> **Why each step is required?** In general:
>
> - The proposed FedTHE learns a federated trained 2-head (global and local) model for each client and adaptively ensembles the 2-head’s prediction via an unsupervised loss $\mathcal{L}_{SLW}$.
> - The unsupervised loss contains an entropy minimization term and a feature alignment term, respectively from prediction and feature space perspectives.
> - Naively minimizing entropy (i.e., assigning larger ensemble weights to a more confident head) results in the 'fake confidence' problem as stated on page 5 middle, but further adding a feature alignment constraint and combining them properly (via similarity-guided loss weighting) would produce a more effective ensemble scheme.
> - In Section 4.2, we demonstrated the importance of each module as well as the uniqueness of our design choice of FedTHE & FedTHE+: as shown in the extensive ablation study of Table 3, we numerically justify why each component is needed and jointly make the solution effective (please also check a detailed analysis in the next response).
>
> **Why can the proposed method be good for FL?** Our goal of this work is to explore **robust personalization for FL** that can improve clean test accuracy while handling **diverse** intra-client distribution shifts, such as corruptions or natural shifts, especially label distribution shift that is more likely to happen in FL scenarios. Filling this gap is crucial, because:
>
> - Previous personalized FL methods only aim for good local clean accuracy;
> - Existing OOD generalization works (such as AFL [1]) for FL only consider tackling inter-client data heterogeneity, leaving intra-client distribution shift unexplored;
> - Naively combing TTA methods with FL fails to produce decent results, especially on label distribution shift (see Table1).
>
> To the best of our knowledge:
>
> - We are the first to identify this problem and achieve consistent improvements on **diverse** types of distribution shifts, while also produces higher clean test accuracy than SOTA baselines. As acknowledged by several reviewers (bhMe, 129X, and you), our method is novel and efficient for FL.
> - Our method is good for FL and simple yet effective: it learns a federated trained 2-head model and uses an unsupervised adaptive ensembling scheme to dynamically combine global and local heads, taking advantage of the collaborative learning rather than directly finetuning on the global model and using fixed personalized models in most personalized FL methods, disturbing the collaboratively trained global model.

---

> > ### Author Response · Authors · 2022-11-11
> > **Response to Reviewer wWXU (2/2)**
> >
> > > Response to the comment: FedTHE, the proposed algorithm, consists of many parts. Many parts have synergy between them, and the synergy produces good performances. However, such a complicated structure makes it difficult to understand which part is the key to dealing with distribution shifts.
> >
> > We would like to kindly clarify that we have an extensive ablation study (Table 3) and the two `remarks` on page 9 give insights on understanding the accuracy trade-off and the design choices. In Table 3:
> >
> > - Combining `row 4&5`, we show that while being orthogonal to our contribution, improving global model quality via Balanced Softmax or other methods such as DRO (mentioned by reviewer bhMe) can consistently improve ID&OOD accuracy.
> > - With `row 5&7`, we demonstrate that entropy minimization (EM), as a way of reducing prediction uncertainty, does not work well under long-tailed local distribution in FL and we hypothesize that it is because the classifier overfits to biased class distribution (as in page 5 middle).
> > - With `row 5&6`, we illustrate that our proposed feature alignment (FA) module is good at handling label distribution shift (Original OoC local test). However, it produces lower accuracy on co-variate distribution shifts (Corrupted and Naturally shifted), indicating that FA cannot handle the case where the feature space is only corrupted rather than switched to the feature space of other classes.
> > - Either EM or FA lead to much better accuracy on label distribution shift case: when tailed or unseen classes show up to a client, the global model is favored by EM or FA. This is because the global model encodes all classes' information, leading to (i) low uncertainty predictions; (ii) better alignment between global descriptor $h_g$ (eq. 4) and test feature $h^{\prime}$. However, as shown in Table 3, neither EM nor FA can provide satisfactory accuracy under co-variate shifts and in-distribution. As our key contribution to dealing with **diverse** distributions (5 distributions in our case), we propose to combine these two modules:
> >     - `Row 8` naively combines them based on the intuition that not only the prediction should have low entropy but also the feature space should be aligned;
> >     - `Row 9` (our final solution) uses an adaptive Similarity-guided Loss Weighting (SLW) strategy following the intuition in page 6, showing decent results while avoiding hyper-parameter tuning overhead.
> >
> > > Response to the comment: Since this paper has too much content with a page limit, many details are omitted in the main body and deferred to the appendix. So it took a lot of work to read.
> >
> > We appreciate your time for reviewing and the feedback! We’ll improve on this point.

---

> ### Author Response · Authors · 2022-11-16
> **Hoping that our response could address your concern**
>
> Dear Reviewer wWXU,
>
> Thank you again for your time and effort in reviewing our work! We would appreciate it if you can let us know if our response has addressed your concern and thus improved your assessment of our paper. We look forward to hearing from you and remain at your disposal for any further clarification that you might require.

---

> > ### Comment · Reviewer_wWXU · 2022-11-17
> > **Thank you for the feedback**
> >
> > I appreciate the feedback from the authors. I've read the authors' responses and all other reviewers' comments. I'm convinced the proposed algorithm is novel and has good accuracy results. However, I still have some concerns about this work. Although there are some ablation studies that explain the importance of each step, it would be great if the authors could provide a more in-depth analysis of the most important part. For instance, visualizations of the feature distributions before and after a module of the proposed algorithm could be useful. The variance with and without moving averages also could be good evidence of this paper's claim.

---

> > > ### Author Response · Authors · 2022-11-18
> > > **Thank you**
> > >
> > > Dear Reviewer wWXU,
> > >
> > > Thank you for the feedback and the acknowledgment of the novelty and effectiveness of our method. We kindly further address your concerns.
> > >
> > > > For instance, visualizations of the feature distributions before and after a module of the proposed algorithm could be useful. The variance with and without moving averages also could be good evidence of this paper's claim.
> > >
> > > We would like to clarify that the test-time robust personalization phase of FedTHE does not tune the feature representation (the output of feature extractor $h_n$) except for adding variance reduction term $h^{history}$ via $h_n^\prime:= \beta h_{n} + (1-\alpha) h^{history}$ in the FA module, so we answer your question of feature distribution based on **the comparison between `with` and `without` $h^{history}$ (moving averages)**.
> > > - We visualized the effect of moving average on head ensemble weight $e$ ([here](https://imgur.com/a/ZMCNrz0)) and $h_n^\prime$ ([here](https://imgur.com/a/HcUGlMd), $h_n$ (w/o history) and $h^\prime$ (w/ history) of a single class test samples visualized). As we can see, when adding the moving average term $h^{history}$:
> > >   - **Analysis of the figure of $e$.** The distribution of ensemble weight $e$ changes in a way that the local head gets a large weight more frequently in the ID case and so does the global head in OoC (label distribution shift) case. This coincides with the intuition that the local head plays a more important role on ID test samples and so does the global head on label distribution shift case (due to its fairness of predicting on all labels).
> > >   - **Analysis of the figure of $h_n^\prime$.** The variance of features can be effectively reduced, leading to a better estimation/calibration of single test feature representation based on the past test samples.
> > >
> > > - On the accuracy side, in Table 3 (`row 9&11`) we provided evidence on *how the variance reduction term $h^{history}$ affects the performance.* When removing the $h^{history}$ term, the ID&OOD accuracy consistently drops, indicating the advantage of adding such a variance reduction term.

---

### Official Review · Reviewer_JEHY · 2022-11-04

**Confidence:** 4
**Correctness:** 3
**Technical Novelty And Significance:** 4
**Empirical Novelty And Significance:** 3
**Recommendation:** 8

**Clarity, Quality, Novelty And Reproducibility:**

Is this method stateful, or stateless (A Field Guide to Federated Optimization Section 7.5 https://arxiv.org/abs/2107.06917)? Is the local head w^l maintained on each client? How many clients in experiments, and is there client sampling for each round?

How is h^{history} computed? Is it intra-client, or inter-client? Why is moving average used, which makes it depending on the order of “history”?

Though I appreciate the extensive results, table 2 is a bit hard to interpret. Maybe I missed it, I would personally see more analysis on e, which is the key of the proposed method. For example, what is the distribution of e for various clients, and in various distribution settings? Why did the proposed method work for various shifts while the previous methods fail.

Both the proposed FedTHE+ and the dataset in section 5 sound somewhat complicated. Would the authors release both dataset and code for reproducibility?


**Strength And Weaknesses:**

Distribution shift/robust personalization is an important problem that only draws attention recently.

The proposed method of interpolation between local and global heads sounds intuitive.

I appreciate the extensive experiments, but also have some concerns on clarity and reproducibility listed below.


**Summary Of The Paper:**

This paper proposed a new personalization method in federated learning: Federated Test-time
Head Ensemble plus tuning (FedTHE+). FedTHE trained a global feature extractor, a global head and local heads for classification, and during the personalization time, a scalar e is learnt to interpolate between predictions by global and local heads.


**Summary Of The Review:**

I would be happy to discuss the clarification questions with the authors.

---

> ### Author Response · Authors · 2022-11-11
> **Response to Reviewer JEHY (1/3)**
>
> Dear Reviewer JEHY,
>
> Thank you very much for the review and feedback, we kindly address your questions as follows.
>
> > Response to the question: Is this method stateful, or stateless? Is the local head w^l maintained on each client? How many clients in experiments, and is there client sampling for each round?
>
> **Stateful or stateless?**: Since most of the SOTA personalized FL methods (except for FedAvg + FT) are stateful, in order to enable fair comparison, in our method the local head $w^l$ is locally kept and maintained during the federated training process, making our algorithm a ‘stateful’ one, as shown in the top part of Figure 3.
>
> However, as claimed in Section 4.2 `Discussion`, we kindly note that the FedTHE does **not** rely on the states and can easily generalize to new clients: when new clients come and receive the global model from the server, they then locally train a local head and apply our test-time robust personalization method for inference.
>
> **Regarding the number of clients & sampling ratio**: As in Appendix C.1 (page 21 middle `Remarks on the number of clients`), we consider 20 & 100 clients for CIFAR10 and 20 clients for ImageNet, and we discussed the reason for our choice there. As for the sampling ratio, upon extensively taking various distribution shifts & data heterogeneity and tuning hyperparameters for each baseline, the client sampling ratio is set to 1 to disentangle the effect of sampling and simplify the experimental setup. As shown below, we further provide experimental results for strong baselines in Table 1, with a low / medium (0.1 / 0.4) sampling ratio, 20 clients, and 100 communication rounds. Our method can still consistently outperform the others. We have added an ablation study on sampling ratio in the appendix for our revised version.
>
> - Sampling Ratio = 0.1 (on CIFAR10 / Dir(0.1))
>
> | Methods | Original local test | Corrupted local test | Original OoC local test | Naturally Shifted test | Mixture of test |
> | --- | --- | --- | --- | --- | --- |
> | FedAvg + FT | 87.13$_{\pm 0.97}$ | 79.72$_{\pm 1.07}$ | 30.92$_{\pm 3.12}$ | 79.50$_{\pm 2.10}$ | 69.32$_{\pm 0.73}$ |
> | FedRoD | 88.23$_{\pm 0.66}$ | 81.35$_{\pm 0.82}$ | 26.80$_{\pm 2.47}$ | 81.47$_{\pm 1.63}$ | 69.46$_{\pm 0.15}$ |
> | MEMO (P) | 88.34$_{\pm 0.54}$ | 81.01$_{\pm 0.27}$ | 27.52$_{\pm 3.27}$ | 81.42$_{\pm 1.66}$ | 69.57$_{\pm 0.50}$ |
> | kNN-Per | 87.29$_{\pm 1.16}$ | 78.30$_{\pm 0.66}$ | 25.90$_{\pm 4.51}$ | 80.62$_{\pm 1.35}$ | 68.03$_{\pm 1.10}$ |
> | FedTHE (Ours) | 88.70$_{\pm 0.71}$ | 81.15$_{\pm 0.44}$ | 53.58$_{\pm 2.69}$ | 80.65$_{\pm 0.69}$ | 70.07$_{\pm 0.93}$ |
> | FedTHE+ (Ours) | **89.48**$_{\pm 0.58}$ | **81.98**$_{\pm 0.66}$ | **53.69**$_{\pm 2.70}$ | **81.66**$_{\pm 0.47}$ | **70.40**$_{\pm 0.69}$ |
>
> - Sampling Ratio = 0.4 (on CIFAR10 / Dir(0.1))
>
> | Methods | Original local test | Corrupted local test | Original OoC local test | Naturally Shifted test | Mixture of test |
> | --- | --- | --- | --- | --- | --- |
> | FedAvg + FT | 87.58$_{\pm 0.77}$ | 80.53$_{\pm  0.96}$ | 32.34$_{\pm  2.39}$ | 80.80$_{\pm  1.62}$ | 70.31$_{\pm  0.32}$ |
> | FedRoD | 88.77$_{\pm 0.53}$ | 81.76$_{\pm  1.11}$ | 27.14$_{\pm  2.5}$ | 81.30$_{\pm  1.27}$ | 69.74$_{\pm  0.11}$ |
> | MEMO (P) | 88.77$_{\pm 0.2}$ | 81.86$_{\pm  0.35}$ | 28.59$_{\pm  2.84}$ | 81.49$_{\pm  1.13}$ | 70.18$_{\pm  0.35}$ |
> | kNN-Per | 88.92$_{\pm  0.67}$ | 80.40$_{\pm  0.37}$ | 30.49$_{\pm  4.52}$ | 81.23$_{\pm  1.43}$ | 70.26$_{\pm  0.72}$ |
> | FedTHE (Ours) | 89.21$_{\pm  0.55}$ | 81.87$_{\pm  1.11}$ | 57.80$_{\pm  3.52}$ | 80.71$_{\pm  1.33}$ | 71.54$_{\pm  0.69}$ |
> | FedTHE+ (Ours) | **89.91**$_{\pm  0.46}$ | **82.16**$_{\pm  1.16}$ | **58.03**$_{\pm  3.42}$ | **81.64**$_{\pm  0.72}$ | **71.79**$_{\pm  0.74}$ |
>
> > Response to the question: How is $h^{history}$ computed? Is it intra-client, or inter-client? Why is moving average used, which makes it depending on the order of “history”?
>
> **Why do we need $h^{history}$?** As opposed to adapting a $e$ for each **batch** of test samples, we choose to design our method by adapting $e$ **per test sample**, which is a favorable property for deployment since online test samples usually come one by one. However, such a strategy naturally introduces high variance in feature representation $h_n$ ($n$ denotes nth test sample) when we cannot have a batch of samples, making feature alignment unstable.
>
> **How is it computed and why use moving average?** As a solution, the $h^{history}$ here is supposed to act as variance reduction term via $h_n^\prime := \beta h_n + (1 − \beta) h^{history}$; it is computed **intra-client** by performing exponential moving average on the past test samples’ feature representations, since we want to **pay more attention to the recent ones**. The history is computationally dependable to test order, whereas we show that it is **not** sensitive to the test order, with the accuracy on **shuffled** ‘mixture of test’ in Table 1 & 2 & 3.

---

> > ### Author Response · Authors · 2022-11-11
> > **Response to Reviewer JEHY (2/3)**
> >
> > **Numerical results for $h^{history}$ and its stabilization effects**: We further show in the ablation study of Table 3 (`row 9` v.s. `row 11`) that this history stabilizes the feature alignment module and plays an important role. Finally, we would like to clarify that the $\alpha$ and $\beta$ hyperparameters for maintaining this process are fixed across all experiments and can generalize between models/datasets/data heterogeneity.
> >
> > > Response to the question: Though I appreciate the extensive results, table 2 is a bit hard to interpret. Maybe I missed it, I would personally see more analysis on e, which is the key of the proposed method. For example, what is the distribution of e for various clients, and in various distribution settings? Why did the proposed method work for various shifts while the previous methods fail.
> >
> > In Table 2, we show experimental results for large-scale ImageNet with its distribution shifted variants, either naturally shifted (ImageNet-V2), style shifted (ImageNet-R) or adversarially shifted (ImageNet-A), and each variant is split to each client according to the client’s class distribution (to avoid inducing label distribution shift). The results show the advancement of our proposed method.
> >
> > **Discussion on head ensemble weight $e$**:
> >
> > - Basically, the scalar ensemble weight is **efficiently** optimized for **each test sample** (Section 1, Section 4 first paragraph and Section 4.2 Test-time Robust Personalization paragraph, etc.) by monitoring prediction entropy and feature alignment of global and local models. We provide more empirical understandings on the choice of $e$.
> > - **Understanding the effect of ensemble weight in Appendix E.4**: in Figure 14 & 15 ($e$ denotes the ensemble weight for the global model), we conduct ablation studies for different ensemble (interpolation) strategies on local and global models. Specifically, we show the accuracy on ‘mixture of test’ with optimal ensemble weight $e$:
> >     - Traversing an optimal **client-shared** ensemble weight (`blue curve`) in Figure 14 and measuring the averaged local accuracy.  The insights are: when ensembling local and global **head** (Figure 14), the optimal $e$ is around 0.8, meaning relying on the local head is beneficial, while in Figure 15 (ensembling local and global **model**), giving higher $e$ to global model is beneficial.
> >     - Traversing an optimal ensemble weight for **each client** and averaging the local accuracy (`red dash line`, the upper bound) in Figure 15. Our method adapts ensemble weight for each test sample, demonstrating better ensembling than the per-client optimal ensemble weight.
> > - **Understanding the ensemble weights under various distribution settings**:
> >     - For **in-distribution** (Original local test, the clean test that is i.i.d to local train), our major observation is that 1) for most samples (major classes), a large weight on the local head is learned since the local head fits local distribution, 2) for a very small portion of samples that belong to tailed local classes, a smaller ensemble weight on local head is learned, in order to better utilize the knowledge from the global head. This is why our method outperforms all baselines on clean test accuracy. Also, a similar observation can be found for **co-variate shifts** (i.e., common corruptions and natural distribution shift), as explained by `Remarks on the ID&OOD accuracy trade-off for FL` (page 9 middle) in Section 6.2.
> >     - For **label distribution shift** (the Original OoC local test column), our major observation (also shown in Table 1) is that the proposed unsupervised adaptive ensembling scheme (FedTHE/FedTHE+) can improve 20%-30% absolute accuracy compared to strong personalized FL baselines, and can achieve accuracy that is relatively close to non-personalized FL methods (e.g. FedAvg). This means that our method assigns larger $e$ (> 0.8) to the global head when encountering tailed or unseen classes, and assigns a larger $e$ to the local head otherwise, doing much better exploitation on the head ensembling.
> >
> > We will further update the manuscript with the above discussion on $e$.
> >
> > **Why did the proposed method work for various shifts while the previous methods fail?** In short, previous methods merely use **fixed** global model (e.g. FedAvg) or personalized model, which inevitably suffers from drastic accuracy drop in test-time distribution shifts (see Figure 2), while our work performs **dynamic** robust personalization (via ensembling weight $e$) according to per test sample, and thus consistently improve the OoD robustness at test-time.

---

> > > ### Author Response · Authors · 2022-11-11
> > > **Response to Reviewer JEHY (3/3)**
> > >
> > > > Response to the question: Both the proposed FedTHE+ and the dataset in section 5 sound somewhat complicated.
> > >
> > > **From the perspective of methodology**, we argue that it is simple & clear (also acknowledged by [Reviewer 129X](https://openreview.net/forum?id=3aBuJEza5sq&noteId=l5ZMyUIu2E)): given a test sample, FedTHE performs test-time adaptive ensembling of local and global head. After obtaining the optimal ensemble weight $e^\star$, we show that other test-time adaptation method naturally fits in, formulating a test-time LP-FT scheme, namely the FedTHE+. We show the advancement of such choice by comparing with naively fine-tuning all parameters (see `row 2` v.s. `row 13` in Table 3).
> > >
> > > **From the perspective of dataset**, we believe the current benchmark properly reflects the overall test-time FL robustness by taking diverse distribution shifts into account, as in Figure 4 and Appendix D. We consider building datasets specialized for FL OoD robustness as a promising direction for the FL community and we leave it to future work.
> > >
> > > To the best of our knowledge, through a test-time adaptation perspective, we are the first to tackle the realistic intra-client distribution shift problem for FL, and contribute a way of evaluation. Our method shows consistent and significant in-distribution and OoD accuracy improvements than other baselines.
> > >
> > > > Response to the question: Would the authors release both dataset and code for reproducibility?
> > >
> > > Yes. The source code is provided as supplementary material, and since our goal is to provide the first personalized FL benchmark and method that handle distribution shifts for the FL community, we will definitely further clean up and release the code and dataset for a larger societal impact.
> > >
> > > We hope the above answers address your concerns, and please don't hesitate to let us know if you have any remaining questions.

---

> > > > ### Comment · Reviewer_JEHY · 2022-11-18
> > > > **Thanks and possibly provide some statistics of e?**
> > > >
> > > > Thanks for the clarification and additional experiments. Most of my concerns are addressed.
> > > >
> > > > Is it possible to provide some details/statistics of e? Or anything that can help justify that e should be optimized for each test sample, not per batch, per client, per dataset. I acknowledge the authors argument that per sample inference is practical, but I guess we can always use some held-out data to determine e.

---

> > > > > ### Author Response · Authors · 2022-11-19
> > > > > **Thank you**
> > > > >
> > > > > Dear Reviewer JEHY,
> > > > >
> > > > > Thank you for the feedback, and we are happy that our response could address most of your concerns. Here we kindly answer your question about $e$.
> > > > >
> > > > > **Is it possible to provide some details/statistics of e?** Yes. Please check our latest revision for the new Appendix E.5 as well as Figure 17, which visualizes $e$ on various distributions. And we kindly provide more evidence as follows.
> > > > >
> > > > > - Why should not $e$ be optimized **per-batch**? As in `row 9 and row 10` of Table 3 for the ablation study, optimizing $e$ for per-batch can give better results on ID and co-variate shift cases while falling short of label distribution shift and mixed test cases. And we would like to clarify that such per-batch optimization on multiple test samples is **not** desirable in the online setting.
> > > > >
> > > > > - Why should not $e$ be optimized **per-client**? In Appendix E.4 `Ablation study for classifier-level interpolation strategies` (page 31) and Figure 14&15, we have a comparison between globally shared $e$, **per-client** oracle $e$ (traverse an optimal $e$ for each client according to accuracy on test set) and our method. We show that our method is on par with **oracle** per-client $e$ case.
> > > > >
> > > > > - Why should not $e$ be optimized **per-dataset**? As in Appendix E.5 and Figure 17 in our latest revision, we see that the per-sample optimized $e$ for a given distribution (dataset) is approximately bimodal. This indicates that using a single optimized $e$ for the whole dataset will give a much worse performance.
> > > > >
> > > > > Thank you for your time and effort that helped us improve the paper, and if our response could address your concern, we appreciate it if you would reconsider your assessment.

---

> > > > > > ### Comment · Reviewer_JEHY · 2022-11-19
> > > > > > **Thanks!**
> > > > > >
> > > > > > Thanks. Figure 17 is what I am interested to see.
> > > > > >
> > > > > > I asked the question because I am trying to understand if there are simpler strategies we can use, as (also pointed out by other reviewers) the method is a little complicated with a lot of moving parts and somewhat ad-hoc design.
> > > > > >
> > > > > > How I interpret Figure 17 is that there are potentially simpler strategies, such as fixing e to a certain value for a specific distribution shift type  (not 100% certain, please correct me if I was wrong), adapting e has an advantage for the mixture of test distribution shown in the last plot in Figure 17.
> > > > > >
> > > > > > I am generally happy with the response, and would raise the score to 8.

---

> ### Author Response · Authors · 2022-11-16
> **Hoping that our response could address your concern**
>
> Dear Reviewer JEHY,
>
> Thank you again for your time and effort in reviewing our work! We would appreciate it if you can let us know if our response has addressed your concern and thus improved your assessment of our paper. We look forward to hearing from you and remain at your disposal for any further clarification that you might require.

---

### Author Response · Authors · 2022-11-11
**To all the reviewers**

We would like to thank all the reviewers for the acknowledgment of the novelty, effectiveness and efficiency of our proposed method, as well as the importance of our proposed test-time robust personalization problem. And we thank all the reviewers for the insightful comments that help us to improve the paper.

In the revised version, we have corrected the bold numbers in Table 1, added [1, 2] to related work, added an ablation study on client sampling ratio in Appendix E.4 (pages 32&33), and improved Figure 14 and 15.

We provided responses to each reviewer separately below.

[1] Deng, Yuyang, Mohammad Mahdi Kamani, and Mehrdad Mahdavi. "Distributionally robust federated averaging." *Advances in Neural Information Processing Systems* 33 (2020): 15111-15122.

[2] Wu, Bingzhe, et al. "DRFLM: Distributionally Robust Federated Learning with Inter-client Noise via Local Mixup." *arXiv preprint arXiv:2204.07742* (2022).

------------
**Update:**
As $e$ is the key of controlling the local-global ensemble, we also updated a visualization and analysis of the head ensemble weight $e$ in Figure 17 and Appendix E.5 (see [here](https://imgur.com/a/K1mr7iL) for a quick look) in the latest revision to better understand how our method decides to ensemble global and local head under different test ID&OOD distributions.

---

### Decision · Program_Chairs · 2023-01-20

**Decision:**

Accept: poster

**Justification For Why Not Higher Score:**

The algorithm contains several parts, and there are insufficient explanations/ablation studies to demonstrate the usefulness of each part.

**Justification For Why Not Lower Score:**

The paper has sufficient novelty and is tackling an important problem.

**Metareview: Summary, Strengths And Weaknesses:**

This paper proposes a new way to personalize the model in federated learning. All the reviewers think that personalization in federated learning is an important problem, and the proposed algorithm is well-motivated. The empirical results also support the claims. Despite having sufficient novelty, the clarity of the paper can be improved. Therefore, we recommend acceptance for this paper.

As suggested by Reviewer wWXU, the paper can be improved by adding more explanations and ablation studies for each step of the proposed method. We hope the authors can address the concerns in the final version.



**Note From Pc:**

if the above contains the word "oral" or "spotlight" please see: "oral" presentation means -> notable-top-5% and "spotlight" means -> notable-top-25%. As stated in our emails, we are disassociating presentation type from AC recommendations